



# Characterization of the mean and extreme Mediterranean cyclones and their variability during the period 1500 BCE to 1850 CE

Onno Doensen[1,2], Martina Messmer[1,2,3], Christoph C. Raible[1,2], and Woon Mi Kim[4,5]

[1]Oeschger Centre for Climate Change Research, University of Bern, Bern, Switzerland
[2]Climate and Environmental Physics, University of Bern, Bern, Switzerland
[3]now at: Delft University of Technology, Civil Engineering and Geosciences, Geoscience and Remote Sensing, Delft, The Netherlands
[4]Climate and Global Dynamics Laboratory, NSF National Center for Atmospheric Research, Boulder CO, USA
[5]now at: Institute of Meteorology and Climate Research, Department of Tropospheric Research, Karlsruhe Institute of Technology, Karlsruhe, Germany

**Correspondence:** Onno Doensen (onno.doensen@unibe.ch)

**Abstract.** Extratropical cyclones are important meteorological phenomena in the Mediterranean and essential for local water supplies, yet they also pose significant hazards for the region as a result of extreme precipitation or wind events. Although they have been extensively studied using global and regional climate models, their variability in the late Holocene is poorly understood. Here, we study a 3350-year climatological simulation that allows us to characterise Mediterranean cyclones better and provides a baseline for more accurately assessing the long-term effects of future climate change on Mediterranean cyclones. To analyse Mediterranean cyclone characteristics, we use a seamless transient simulation from 1500 BCE to 1850 CE produced by the Community Earth System Model with a 6-hourly temporal and $1.9° \times 2.5°$ horizontal resolutions. We found that Mediterranean cyclones exhibit pronounced multi-decadal variability in the order of 5% throughout the entire late Holocene. For the cyclone frequency, a relationship is identified with the East-Atlantic, the East-Atlantic Western-Russia, and the Scandinavian modes of circulation. Cyclone frequency shows, although small, a significant increase in the eastern Mediterranean after severe volcanic eruptions with the highest sulphur injections. The composites of the most extreme cyclones with respect to wind speed and precipitation indicate that cyclones in the central Mediterranean have the potential to grow more intense over their entire lifetime than cyclones in the eastern Mediterranean. This is especially true for cyclones with extreme wind speed, implying that people in the central Mediterranean are potentially more exposed to hazards caused by extreme cyclones.

## 1 Introduction

The Mediterranean has long been recognized as one of the active storm track regions in the world. Due to its uniquely complex topography, Mediterranean cyclones are generally smaller in scale, less intense, and of shorter life span than cyclones in other major storm track areas (Trigo, 2006; Flaounas et al., 2014). Yet, these cyclones can heavily impact the Mediterranean region as a result of heavy precipitation events (Pfahl and Wernli, 2012; Flaounas et al., 2015a; Raveh-Rubin and Wernli, 2015), heavy wind storms (Nissen et al., 2010; Raveh-Rubin and Wernli, 2015), and coastal floods (Lionello et al., 2019; Ferrarin et al., 2021). These hazards can also occur simultaneously, resulting in compounding events that can have larger impacts than





individual events combined (Zscheischler et al., 2018; Messmer and Simmonds, 2021; Vakrat and Hochman, 2023; Portal et al., 2024). Nonetheless, Mediterranean cyclones are important for local water and energy supplies due to the precipitation they bring into this semi-arid region (Seager et al., 2014). However, the factors driving its variability, as well as its connection to extreme cyclones, remain unclear. Thus, the purpose of this study is to deepen our understanding of Mediterranean cyclones by analysing past climate variability effects on the frequency and intensity of these types of cyclones using the Community Earth System Model (CESM; Hurrell et al., 2013).

Mediterranean cyclones occur more frequently in winter and have a frequency minimum in summer, with extreme cyclones being even less likely in summer (Flaounas et al., 2015b). The main cyclone hotspots in the Mediterranean are over the Gulf of Genoa, the Adriatic Sea, the Tyrrhenian Sea, the Aegean Sea, and around Cyprus (Lionello et al., 2016; Flaounas et al., 2018). Most Mediterranean cyclones originate in the Mediterranean basin, with only about 20% of cyclones developing over an area outside the Mediterranean (mainly the Atlantic) (Lionello et al., 2016). Within the Mediterranean, several subtypes of cyclones are classified. A well-known subtype is the Genoa low that forms over the Gulf of Genoa and can heavily impact weather over the Alps, southern France, and Italy, often being strengthened by lee effects caused by the Alps (Mesinger and Strickler, 1982). Sometimes, these Genoa lows follow a track northeastwards into central Europe and are then classified as Vb-cyclones (van Bebber, 1891). These are quite rare events, but can heavily impact the Alps and central Europe as they often cause heavy precipitation (Mudelsee et al., 2004; Messmer et al., 2015; Krug et al., 2021; Stucki et al., 2020). Another type of lee cyclones is the so-called Sharav cyclone, which usually forms in spring on the lee side of the Atlas Mountains and travels eastwards rapidly. Due to their origin in arid areas, they often contain little moisture and can trigger dust storms over the Mediterranean (Alpert and Ziv, 1989). A special type of Mediterranean cyclone is the so-called Medicane, which is a hybrid system between tropical and extratropical storms and often occurs in autumn. Medicanes have received lots of attention in recent years due to their intensity, however, their overall socio-economic impact is limited due to their rarity (Flaounas et al., 2022).

Extratropical cyclone activity, in general, shows a distinct decadal variability (Feser et al., 2015), and this variability can partly be explained by variations in different atmospheric modes of circulation (Flaounas et al., 2022). The most important mode of circulation influencing Mediterranean cyclones is the North Atlantic Oscillation (NAO). The frequency of cyclones in the Mediterranean correlates inversely with the NAO phase (Raible et al., 2007) and is also inversely correlated to wintertime precipitation over the Mediterranean (Brandimarte et al., 2011; Montaldo and Sarigu, 2017). In general, a positive NAO phase is correlated with a decrease in cyclone frequency over the Mediterranean, whereas an increase in cyclone frequency is observed during the negative phase of the NAO. Hofstätter and Blöschl (2019) showed that clustering of Vb-cyclones becomes more frequent during negative NAO-phases. In addition to the NAO, the East Atlantic (EA), Scandinavian (SCAN) and East Atlantic Western Russian (EAWR) and the Polar-Eurasian (POL) modes of variability have a significant influence on cyclones in the Northern Hemisphere (Seierstad et al., 2007). In addition to the modes of atmospheric circulation, Walz et al. (2018) hypothesized that sea ice anomalies are a dominant factor in the inter-annual variability of Mediterranean cyclones.

Besides natural influences, there is evidence that anthropogenic climate change has altered and is expected further to affect the frequency and intensity of Mediterranean cyclones. Winter precipitation in the Mediterranean declined during the 20th century, and one of the causes is a decrease in cyclogenesis in the region (Trigo et al., 2000). In addition to that, Nissen et al.



(2014) suggested that NAO values may get more positive in the future, implying a northward shift of the jet stream, which will reduce cyclone activity in the Mediterranean. An increase in pressure (Hochman et al., 2020), a decrease in baroclinicity, and an increase in static stability (Raible et al., 2010) are further effects of climate change decreasing the number of cyclones in

the Mediterranean.

General circulation models (GCM) are extensively used to analyse the future impact of climate change and have also been extensively used to analyse Mediterranean cyclones and their future changes (Raible et al., 2010; Nissen et al., 2014). However, GCMs often have systematic biases related to extratropical cyclone frequency and intensity because of the low horizontal resolution and biases in land and sea surface temperatures (Zappa et al., 2013; Priestley et al., 2023). Mediterranean cyclones,

which are usually of a smaller scale than other extratropical cyclones, suffer even more from the low resolution in GCMs (Flaounas et al., 2013). Moreover, the relatively short period that simulations from GCMs cover makes analyses of the long-term variability of extreme cyclones more difficult (Raible et al., 2018). Hence, large uncertainties remain for future projections concerning extreme Mediterranean cyclones.

An alternative method is to look at proxies to reconstruct the climate of the past (Xoplaki et al., 2003). Yet for cyclones,

it is complicated to use proxies since they are usually only sensitive to temperature and precipitation and not to wind and pressure (Raible et al., 2021). Cyclone-related proxies often have to be derived from alterations in dust transport (Jong et al., 2006) or changes in sea sediment depositions (Degeai et al., 2015; Pouzet and Maanan, 2020). Still, some proxy-based records show that periods with more and less storm activity alternate during the Holocene (Degeai et al., 2015; Sabatier et al., 2012). Alternatively, reconstructions of the NAO can be used as an indirect measure of cyclone activity (Pinto and Raible, 2012;

Ortega et al., 2015).

To overcome these problems, paleoclimate modelling that simulates much longer time periods than most simulations generally do is used (Raible et al., 2021). Thereby, GCMs attempt to reproduce past climates by considering natural and external forcings and boundary conditions. The resulting simulations of the past climate can be verified by comparing them against proxies (2k PMIP3 group, 2015; PAGES Hydro2k Consortium, 2017). Simulations of the past climate are also used to evaluate

the ability of GCMs to represent internal climate variability, as well as create a climatological baseline for specific events (Harrison et al., 2015; Kageyama et al., 2018). Several studies have looked at the effect of climate variability during the Holocene on extratropical cyclones in the Northern Hemisphere (Raible et al., 2021; Varma et al., 2012; Xia et al., 2016), or on a synoptic scale in the Euro-Atlantic domain using GCMs (Raible et al., 2018). Raible et al. (2018) simulated the period between 850 CE and 2100 CE and showed that cyclones in the North Atlantic exhibit pronounced low-frequency variations caused by internal

variability of the climate system. A similar finding was also reported by Gagen et al. (2016) for summer extratropical cyclones in the North Atlantic. Regarding natural forcing, Andreasen et al. (2024) suggested that strong volcanic eruptions may increase the number of cyclones in the subtropics and high latitudes, whereas the number of cyclones in the midlatitudes decreases. However, there are only a few studies available on the effects of volcanic eruptions on extratropical cyclones.

In this study, we use the CESM to investigate the effects of past climate variability on the frequency and intensity of Mediter-

ranean cyclones. The CESM simulation consists of a single-member transient run that covers the period from 1500 BCE until 2100 CE (3600 years in total) (Kim et al., 2021). We track all cyclones in the Mediterranean and its immediate surroundings





until 1850 CE. We particularly focus on extreme Mediterranean cyclones for precipitation and wind speed. This approach aids in assessing and understanding the underlying processes that affect (extreme) Mediterranean cyclones and making better projections for the future.

The paper is organized as follows: In Sect. 2, we introduce the CESM simulation and tracking algorithm for Mediterranean cyclones in more detail. In Sect. 3, we evaluate the CESM simulations, show the internal climate variability and atmospheric modes of circulation related to extratropical cyclones, study the impact of volcanic eruptions on extratropical cyclones and show typical cyclone characteristics for different extremes for the central and eastern Mediterranean produced by CESM. Finally, in Sect. 4, we discuss the impacts of our results in a wider framework.

## 2   Methods and Data

### 2.1   General circulation model and reanalysis data

The CESM version 1.2.2 is a fully coupled global climate model consisting of various component models. We used a configuration with a horizontal resolution of $1.9° \times 2.5°$ in the atmosphere and over land, and with a nominal $1.0° \times 1.0°$ resolution for the ocean and sea ice. The atmospheric component contains 30 vertical sigma pressure levels, and the ocean has 60 unevenly
distributed layers. We used a seamless simulation spanning from 1500 BCE to 2012 CE with a 6-hourly temporal resolution, branched off a spin-up equilibrated simulation with 1501 BCE conditions (Kim et al., 2021). The simulation was forced by solar irradiance, greenhouse gas (GHG) concentrations, albedo changes, and volcanic sulphur injections into the stratosphere that covers the entire simulation period (Fig.1). Most of the forcings were obtained from the Paleoclimate Modelling Intercomparison Project phase 4 (PMIP4) database, and the volcanic forcing was from Sigl et al. (2021) and Carn et al. (2016).
How these forcings were obtained and generated are explained in detail Kim et al. (2021). In this study, we used geopotential at 1000 hPa (Z1000) and 500 hPa (Z500), sea level pressure (SLP), 850 hPa temperature anomaly (T850), total precipitation, which is the sum of the large-scale and convective precipitation, and horizontal wind components at 850 hPa and 300 hPa in the $x$- and $y$-direction, which are combined to wind speeds WS850 and WS300, respectively.



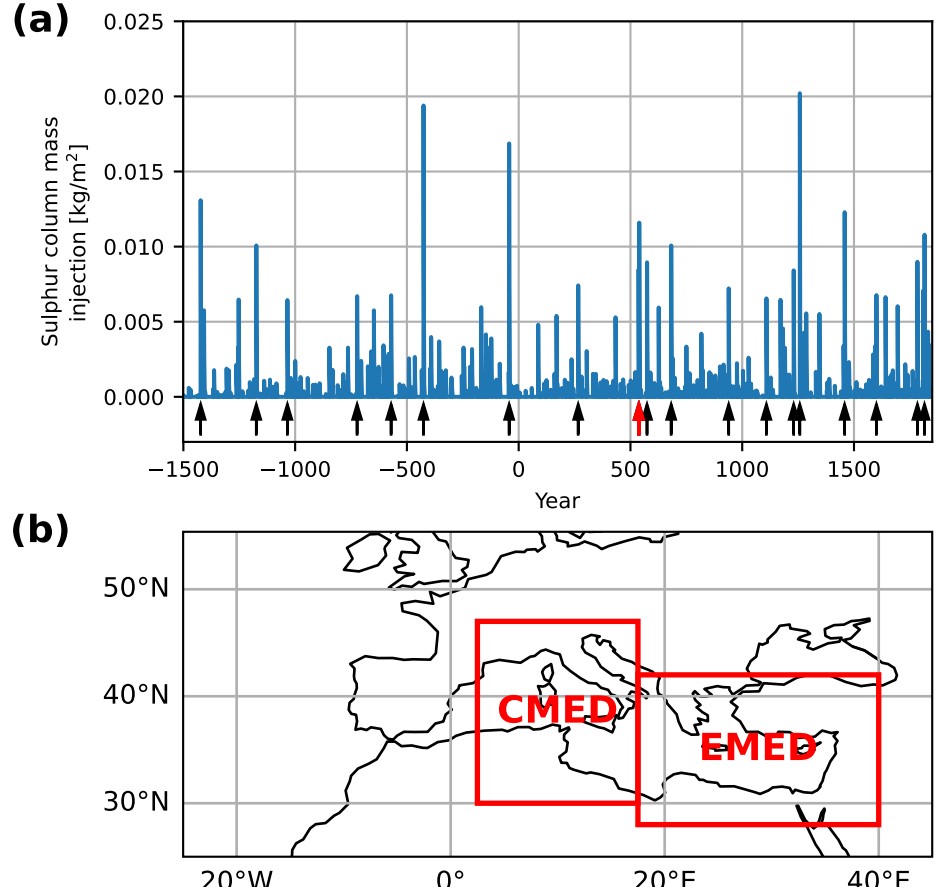

**Figure 1.** (a) Overview of stratospheric sulphur injections into the stratosphere as a result of volcanic eruptions. The arrows indicate when the 20 eruptions with the highest sulphur injections occurred. The red arrow indicates two top 20 eruptions occurring in quick succession. (b) The two regions of interest are shown: "CMED" indicates the region defined as the central Mediterranean, "EMED" indicates the region defined as the eastern Mediterranean.

To evaluate the representation of Mediterranean cyclones in CESM, we compared the CESM output with the ERA5 reanaly-
sis (Hersbach et al., 2020) for the period where both datasets overlap in time, which is from 1981 to 2010. ERA5 was regridded to the same spatial resolution as CESM to provide the fairest possible comparison between the two.

## 2.2    Cyclone detection and tracking algorithm

We applied the cyclone detection and tracking algorithm developed by Blender et al. (1997) and extended by Schneidereit et al. (2010) to the seamless model simulation and ERA5 data. The algorithm identified low-pressure systems in the Z1000 field. To
exclude weak and unrealistic low-pressure systems, the cyclones need to fulfil the following requirements:



- A minimum gradient of at least 20 geopotential meters (gpm) per 1000 km is achieved.

- During the whole cyclone's lifetime, a minimum gradient of 30 gpm per 1000 km is achieved at least once.

- Local minima over grid cells with an orography higher than 1000 m above sea level are excluded.

- To connect different minima to a cyclone track, a minimum in the following time step is identified with a next-neighbour search. The new minimum of the cyclone must be within 1000 km of the previous cyclone minimum across 24 hours.

- The minimal lifetime of a cyclone is at least 24 hours.

The cyclone identification and tracking method provided a variety of cyclone characteristics, such as the cyclone position, the radius of a cyclone, the cyclone depth, the core (central) pressure, and cyclone-related mean and extreme precipitation and wind speed. To compute the cyclone radius, first, a Gaussian function was fitted to the Z1000 field, assuming that the cyclone was azimuthally symmetric (Schneidereit et al., 2010). The cyclone radius is then defined as the distance between the first inflection point and the cyclone centre (which represents one standard deviation). The depth of the cyclone is defined as the difference between Z1000 in the centre of the cyclone and the Z1000 mean over the area of $1000 \times 1000 \ km^2$.

To calculate the cyclone-related wind speed and precipitation, the maximum value of wind speed and precipitation of all grid cells within a cyclone's radius were considered Raible et al. (2018). Although radii computed by the algorithm for synoptic-scale cyclones of the Atlantic are quite realistic, for the Mediterranean, the computed radii were generally too small, resulting in a cyclone area of one grid point. Since the traditional selection of one standard deviation of the Gaussian fit to the geopotential height field as the definition of the radius was to some extent subjective, we adjusted the radii from the tracking algorithm by a factor of 1.5. With this adjustment, we transformed the radius to resemble the point where the Laplacian of the geopotential height field is equal to zero (Messmer and Simmonds, 2021). We also applied the tracking algorithm to ERA5 Z1000 field to evaluate the Mediterranean cyclone characteristics in CESM, mostly focusing on the cyclone tracks.

## 2.3 Superposed epoch analysis

To test the potential impact of volcanic eruptions on the frequency of Mediterranean cyclones, we applied the superposed epoch analysis on the simulation by analysing the mean conditions before and after the 20 strongest eruptions (Lehner et al., 2015; Raible et al., 2018). We considered the conditions of the five years before an eruption occurs to be the baseline and compared the conditions of the two years after the eruption as the perturbed comparison state. If two or more eruptions occurred within five years of each other, we used the first five unperturbed years occurring before these eruptions as the unperturbed state and the two years after the last eruption as the perturbed state. To identify whether the cyclone frequency in the period after the eruption is significantly different to before the eruption, we apply a Welch's t-test (Welch, 1947) using 5% signficance level.

## 2.4 Atmospheric modes of circulation and Mediterranean cyclones

To find the relation between atmospheric modes of circulation and Mediterranean cyclones, we first computed the most dominant modes of circulation over the North Atlantic domain in the CESM simulation. We applied an empirical orthogonal





function (EOF) analysis to anomalies of Z500. For that, a 100-year period was selected from the simulation to compute the most important modes on seasonal time scales, and then these modes were projected on the rest of the monthly Z500 data for the whole simulation until 1850. The EOF analysis was calculated for the region between 90° W - 40° E and 20° N - 80° N, similar to Hurrell (1995). Finally, the principal components (PCs) of the EOFs were computed and normalized. To identify a relation between atmospheric modes of circulation and Mediterranean cyclones, the Pearson correlation coefficients were estimated between the PCs and the monthly cyclone frequency, cyclone-related precipitation and cyclone-related wind speed. We only considered grid cells where the model orography is below 1000 m and where a cyclone frequency was higher than 0.01 $day^-1$.

## 2.5 Area of interest and analysis of extreme Mediterranean cyclones

Mediterranean cyclones are subject to large temporal and spatial variability (Campins et al., 2011). Therefore, we split the Mediterranean region into the central and the eastern Mediterranean (Fig. 1) and analysed extreme cyclones for each of these regions. We exclude the western Mediterranean Sea, as there were only a very limited number of grid points available for this region due to the rather coarse resolution of CESM, and based on our own analysis the western Mediterranean is heavily influenced by the Atlantic, and therefore the dominant processes here are not so relevant for the rest of the Mediterranean.

To analyse extreme cyclones, we need to select the most extreme cyclones within the tail of their probabilistic distribution. To do this, we ranked them based on the maximum 6-hourly cyclone-related precipitation and 850 hPa wind speed at the time when the cyclone reaches the lowest core sea level pressure within its track (hereafter **t0**). The location of minimum pressure is obtained while the cyclone track stayed within the respective region of interest, which is either the CMED or EMED (compare Fig. 1), but the track can start and end outside this area. For the two regions, we estimated the distributions of the 6-hourly cyclone-related precipitation and 850 hPa wind speed at **t0** for all cyclones within the corresponding region. Since the distribution for precipitation is skewed towards extreme values, all 6-hourly precipitation rates below 1 mm were excluded and the square root of the untransformed distribution was taken. Compound extreme precipitation and extreme wind events related to cyclones were defined with the joint probability distribution of the 6-hourly cyclone-related precipitation and the 850 hPa wind speed of all cyclones. Next, all the cyclones were ranked in terms of extremeness based on these distributions. We selected the 10, 100, and 1000 most extreme cyclones (hereafter EXC10, EXC100, EXC1000, EXC is used for extreme cyclones in general) for the central and eastern Mediterranean with respect to precipitation, wind speed, and compound events.

Finally, a composite analysis was performed on the EXC10, EXC100, and EXC1000 to investigate the spatial and temporal characteristics of cyclones associated with an extreme event. The following steps are carried out:

1. For each cyclone track, we set a reference time **t0** as the time $t$ with a minimum sea level pressure value within the track while the cyclone stays within the respective domain. Every time step of the track that occurred after **t0** obtains a positive index, and the time steps of the track that occurred before **t0** receive a negative index.

2. For each of the 6-hourly time steps of the cyclone's track, all fields have been centred at the location of the cyclones, given by its core minimum pressure. With this approach, the model data for each cyclone track point was independent





185 of its geographical location. Since all the cyclones were located in the Mediterranean basin **t0** and, therefore, within a relatively small range of latitudes, variations due to differences in latitude weights were considered insignificant and, therefore, negligible.

3. We were only interested in the 30 hours (5-time steps) before and after **t0** each cyclone to capture the intensification and decaying phase of the cyclone.

190 4. For every time step within these 30 hours before and after **t0**, we compute the average of all the fields to gain a spatial average for EXC10, EXC100, and EXC1000. This was done for both regions for cyclones associated with cyclone-related precipitation, wind, and compound extreme. Cyclone tracks, which did not appear at any of these time steps, were ignored for the temporal means.

The composite analysis was applied to the 6-hourly precipitation rate, WS300, WS850, SLP, T850, Z500, and the Rossby
195 wave packed (RWP) amplitude, which is based on Fragkoulidis et al. (2018). A Welch's t-test was applied to all the temporal means and the spatial means of 6-hourly precipitation and WS850 between the two regions to assess whether EXCs between the two regions are significantly different.

## 3 Results

### 3.1 Evaluation of the CESM simulation against ERA5

200 In Fig. 2, we show the cyclone frequency for the period 1981–2010 in CESM and ERA5. Considering Europe and the North Atlantic, CESM reproduces the general storm tracks stretching from Newfoundland, via Iceland towards Scandinavia. However, CESM overestimates cyclone frequency over the polar North Atlantic and underestimates cyclone frequency in the subtropical North Atlantic. This bias is present in all seasons, but is particularly present during autumn (SON) and winter (DJF), with differences up to 50%. Thus, the biases generally indicate that storm tracks in CESM are too zonal compared to wavier storm
205 tracks in ERA5.



**Figure 2.** Cyclone frequency [$day^{-1}$] defined as the number of times a grid cell is located within the calculated radius of a cyclone during the respective season, divided by the number of days of the season in CESM (left column), ERA5 (middle column) and the difference between CESM and ERA5 (right column) for spring (MAM; first row), summer (JJA; second row), autumn (SON; third row) and winter (DJF; fourth row) for the time period 1981–2020. The gray shading indicates areas where the model elevation is more than 1000 metres above sea level.

On a more regional scale, some noticeable biases are visible. First, CESM strongly overestimates cyclone frequency west of Greenland for all seasons. These signals can mostly be ignored, since most of them are artefacts due to the presence of the Greenland ice sheet. Secondly, in the summer months, the number of cyclones over southern Europe is overestimated, especially over the Iberian Peninsula. This is most likely caused by an unrealistic amount of heat lows in CESM. In the central Mediterranean, the cyclone frequency is underestimated by approximately 50% in winter. This could be related to the fact that the storm tracks in CESM are too zonal, and therefore cyclones from the Atlantic penetrate the Mediterranean too little. The latter would also hamper lee cyclogenesis in the Mediterranean, further explaining the underestimation of cyclone frequency.



However, the slight overestimation in the eastern Mediterranean suggests that the zonality in CESM only plays a role in the central Mediterranean.

To assess the representation of cyclone characteristics in CESM, we compare several different cyclone-related variables in CESM and ERA5 for the Mediterranean in Fig. 3. CESM slightly overestimates the cyclone lifetime (Fig. 3a–d), although the number of very short-lived cyclones is underestimated in CESM. Maximum cyclone depth is represented well, although it is slightly underestimated in CESM (Fig. 3e–h), apart from JJA where it is generally overestimated. CESM can reproduce the core minimum pressure for DJF, MAM, and SON (Fig. 3i–l), showing consistent peaks of maximum and minimum frequencies compared to ERA5. However, larger deviations between CESM and ERA5 are found for JJA where the peak of the distribution is about 10 hPa higher for CESM compared to ERA5. This is most likely due to the heat-low bias we observe for JJA in Fig. 2d. Larger biases occur for cyclone-related maximum precipitation (Fig. 3m–p) and cyclone-related maximum wind speed (Fig. 3q–t). CESM struggles to reproduce high cyclone-related precipitation events in all four seasons, where precipitation is most often underestimated by 50%. There is also an underestimation of cyclone-related high wind speed events in all four seasons, but this bias is less strong than the precipitation bias. Nevertheless, the fact that CESM mostly agrees with ERA5 on cyclone position and strength gives us confidence in the model's representation of cyclones over the Mediterranean.





**Figure 3.** Histograms of Mediterranean cyclone-related features in CESM (blue) and ERA5 (orange) between 1981 and 2010 in MAM (first row), JJA (second row), SON (third row), DJF (fourth row). Shown are the relative frequencies of the total lifetime of the cyclone tracks in days (a–d), maximum cyclone depth during its lifetime in gpm (e–h), minimum sea level pressure during its lifetime in hPa (i–l), Maximum cyclone-related precipitation rate during its lifetime in mm/6h (m–p) and maximum cyclone-related wind speed during its lifetime in ms$^{-1}$) (q–t).

## 3.2 Internal climate variability in CESM

Fig. 4 shows the long-term climate variability over 3350 years (1500 BCE to 1850 CE) represented by 30-year running means of T850 and several cyclone-related features. The anomalies are with respect to the averages over the entire 3350-year period.

We consider the 30-year running mean of T850 as a proxy for the state of the climate from 1500 BCE to 1850 CE. In general, pronounced multi-decadal variability is detected in Fig. 4. Warmer and colder periods alternate, with maximum anomalies in the order of 0.5 °C. The period between 1200 BCE and 500 BCE is dominated by warmer conditions, whereas the period between 1600 CE and 1850 CE is dominated by colder conditions also known as the Little Ice Age.



The cyclone frequency in the Mediterranean (Fig. 4b) exhibits a similar pattern to the temperature anomalies (Fig. 4a). The

cyclone frequency in this figure is defined as the total number of grid cells that are occupied by a cyclone centre every month. Cyclone frequency exhibits a multi-decadal variability that varies in the order of 5%. At the beginning of the simulation, a weak tendency towards a higher number of cyclones is found, whereas, towards the end of the simulation, there is a tendency towards fewer cyclones.

**Figure 4.** 30-year running means of air temperature and several cyclone-related properties in the central and eastern Mediterranean regions during 1500 BCE and 1850 CE. Shown are (a) the 850 hPa temperature anomalies in [°C], (b) cyclone frequency in $month^{-1}$, (c) the 90th percentile of maximum cyclone-related precipitation in mm, (d) the 90th percentile of maximum cyclone-related wind speed in ms[-1]

In Fig. 4c and 4d, we present the 30-year running mean of the 90th percentile of monthly maximum cyclone-related pre-

cipitation and cyclone-related wind speed, respectively. Same as the other cyclone-related features, these time series are also dominated by multi-decadal variability. Periods with cyclones that achieve higher precipitation rates and wind speeds often coincide (Table S1). However, these variations are not driven by temperature anomalies or solar irradiance variations (not shown).





### 3.3 Impact of volcanic eruptions on cyclones

To assess the impact of volcanic impact on extratropical cyclones, we show the mean winter cyclone frequency five years before the 20 strongest eruptions in Fig. 5a, and the mean change in cyclone frequency within two years after these eruptions (Fig. 5b, with stippling showing whether the difference is statistically significant. The mean cyclone frequency of the undisturbed (no volcanic eruptions) years (Fig. 5a) resemble the pattern of the mean winter cyclone frequency of all years, as the comparison with Fig. 2j shows.

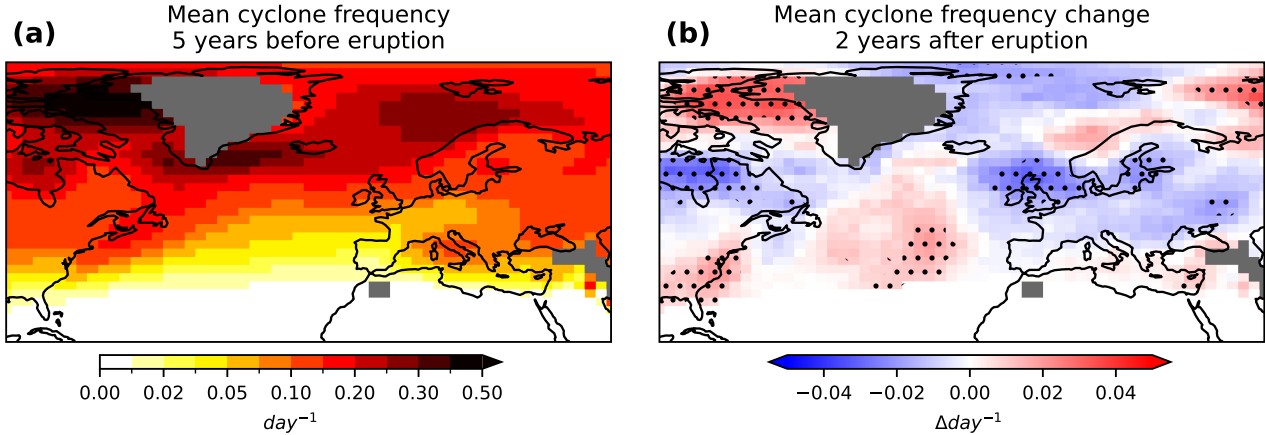

**Figure 5.** The mean cyclone frequency over northern Europe and the North Atlantic 5 years before the 20 strongest eruptions (a), and the changes in mean cyclone frequency that occur two years after the strongest eruptions (b). The stippling indicates where the changes are statistically significant (p=0.05). All results are for DJF.

The changes in cyclone frequency show a clear impact of volcanic eruptions (Fig. 5 b). Over most of Europe, north-eastern North America, and the Arctic Sea, the number of cyclones decreases after a strong volcanic eruption. However, these decreases are only significant in the North Sea, Baltic Sea, and the Hudson Bay. We identify significant cyclone frequency increases over the southeastern US, the subtropical Atlantic, the Barents Sea, and the eastern Mediterranean. Due to the aforementioned biases in Fig. 2 around the Greenland ice sheet, we ignore the large frequency increase west of Greenland. Although significant
changes are evident in many regions of Europe, the Atlantic, and North America, only a small significant increase in cyclone frequency is found in the eastern Mediterranean. Still, this seemingly small increase is in the order of 30% for this region, though.

### 3.4 Atmospheric modes of circulation and cyclone features

Since the large-scale variability can be an import driver for Mediterranean cyclones, we show the impact of the four most
dominant modes of circulation in the North Atlantic European region, resulting from the EOF-analysis applied to the CESM Z500 fields, on extratropical cyclones in Fig. 6. The observed patterns have strong similarities to the North Atlantic Oscillation



(NAO), East Atlantic pattern (EA), East Atlantic Western Russia pattern (EAWR), and the Scandinavian pattern (SCAN) (Fig. S1). The NAO explains most of the variability in Z500 in DJF with 26.8 % (Fig. 6), which is in agreement with previous studies (i.e. Barnston and Livezey (1987)). The NAO is followed by the SCAN-pattern (11.8 %), the EA-pattern (10.8 %), and

the EAWR-pattern (9.6 %).

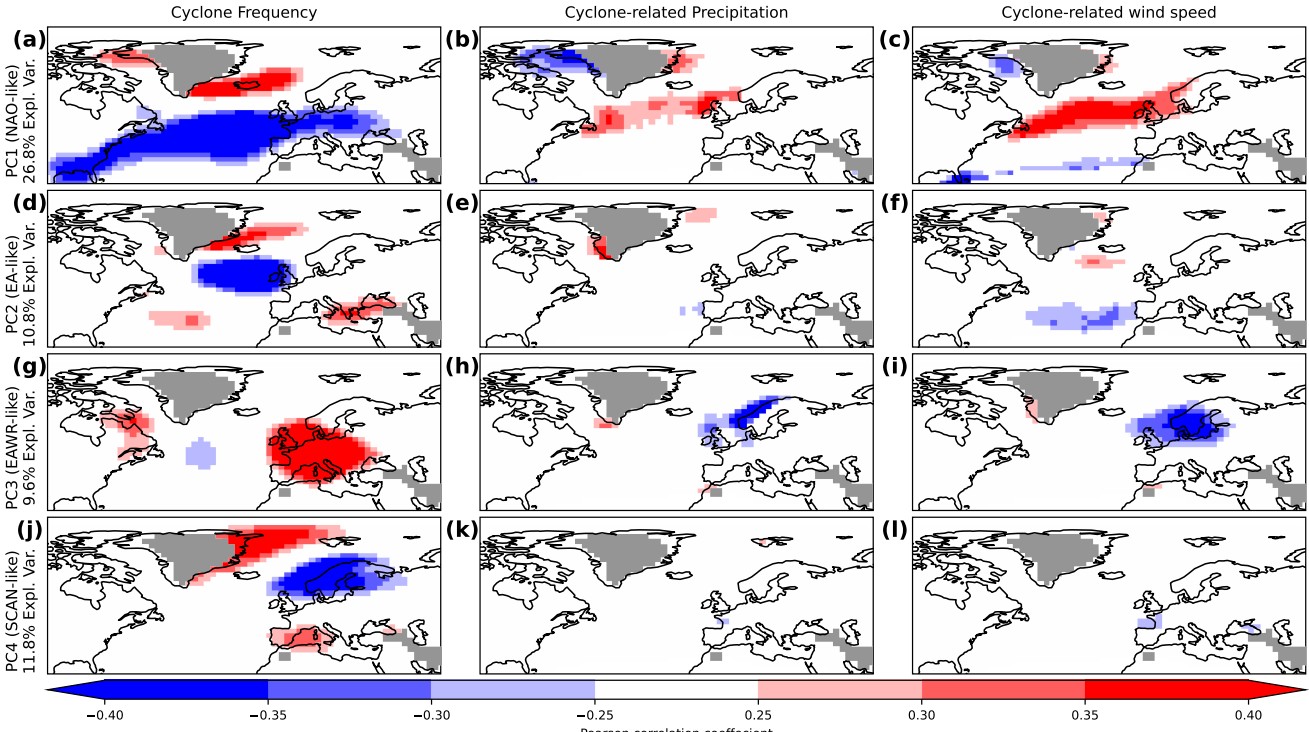

**Figure 6.** The principal components that represent the (a–c) North Atlantic Oscillation (NAO), (d–f) East Atlantic pattern (EA), (g–i) East Atlantic Western Russia pattern (EAWR), and (j–l) Scandinavian pattern (SCAN) correlated to the local cyclone frequency in $day^{-1}$ (left column), maximum cyclone-related precipitation in mm/6h (middle column), and maximum cyclone-related wind speed in ms⁻¹ (right column), using the Pearson correlation analysis. Red shading indicates a positive correlation, whereas blue colours indicate a negative correlation. The grey colours indicate grid cells with model topography above 1000 m. The labels on the y-axis indicate the explained variance for each mode of circulation. All results are for DJF. The mean structures of the different patterns are shown in Supplementary Figure Fig S1.

Fig. 6 shows the correlation coefficients between the modes of circulation and several cyclone features. Generally, the strongest correlations between the four dominant atmospheric modes and cyclone-related features are found over the North Atlantic. The NAO is the strongest driver of cyclone features over the North Atlantic and northern Europe (Fig. 6a-c). A strong negative correlation is found between the NAO and cyclone frequency over the Atlantic between 30° N and 50° N (Fig. 6a),

implying a northward shift of the jet when the NAO is in a positive state, and vice versa. Also, the NAO is positively correlated





with cyclone-related precipitation and wind speed in the typical storm track regions (Fig. 6b-c). In contrast, the subtropical North Atlantic shows a slight tendency for lower cyclone-related wind speeds when the NAO is in a positive state.

The EA phase is less impactful than the NAO on cyclone features (Fig. 6d-f). A positive EA phase is related to a decrease in cyclone frequency over the North Atlantic, whereas the area north of Iceland exhibits an increase in cyclone frequency. There is also an indication that cyclone frequency increases in the subtropical Atlantic and the eastern Mediterranean during a positive EA phase, and vice versa during a negative phase. The relationships between cyclone-related precipitation in Fig. 6e can be mostly ignored due to the biases in CESM close to Greenland. A weak relation between cyclone-related wind speed and the EA phase is identified around the Azores, and an even weaker relation around Iceland.

The EAWR phase (Fig. 6g-i) exhibits a strong positive correlation with cyclone frequency over central Europe and the central Mediterranean, where a positive EAWR phase leads to an increase of cyclones in this area. Despite a strong positive correlation, EAWR is not significantly correlated with the cyclone-related precipitation and wind speed over this region. A positive EAWR phase is related to a decrease in cyclone-related precipitation around Great Britain and the Norwegian coast, as well as a decrease in cyclone-related wind speed over northern Europe.

Lastly, the phase of the SCAN pattern (Fig. 6j-l) only has an impact on the cyclone frequency. A SCAN phase is negatively correlated with a decreased number of cyclones over Scandinavia, and positively correlated with the number of cyclones in the Arctic Sea and the western Mediterranean.

In contrast to the results of Walz et al. (2018), no significant correlation between the sea ice anomalies and cyclone characteristics are found. Therefore, we do not show the results.

For JJA, we find very similar patterns to those during DJF for the correlation between the atmospheric modes and cyclone features (Fig. S2). The main differences to DJF are that the atmospheric modes are less dominant due to the more chaotic nature of the atmosphere in summer and that, therefore, the correlations are less strong. This is especially true for the correlations involving cyclone-related precipitation and wind speed.

In summary, there is a substantial signal for a relation between the dominant atmospheric modes and cyclone features in the North Atlantic and northern Europe, especially in DJF. However, the only signal that appears for the Mediterranean is a weak relation between the phase of the EA, EAWR, and SCAN pattern and cyclone frequency in Fig. 6. Hardly any relationship appears between modes of variability and cyclone-related precipitation and cyclone-related wind speed over the Mediterranean.





### 3.5 Composites of extreme Mediterranean cyclones

**100 most extreme cyclones in the Mediterranean**

**Figure 7.** Composite analysis for the 100 most extreme cyclones (EXC100) for the period 1500 BCE–1850 CE in the central Mediterranean (a–c and g–i), and the eastern Mediterranean (d–f and j–l) at the time when the cyclone achieves minimal sea level pressure (**t0**). The EXC100 are estimated with respect to precipitation (top row), wind speed (middle row), and compounding events (bottom row). In panels a–f, the colour shading shows WS850, and in panels g–l, it indicates 6-hourly precipitation. The hatching indicates where the difference for one event type is statistically significant between the two regions (p=0.05). The grey contours indicate the composite mean sea level pressure.

To characterize extreme Mediterranean cyclones, we use a composite analysis. In Fig. 7a–f, we show WS850 and the 6-hourly precipitation composites for the EXC100 for the period 1500 BCE–1850 CE during DJF. Although these composites



correspond to three distinct EXC types, it is important to acknowledge that an individual EXC may appear within the top 100 rankings across multiple metrics (Table S2).

In the composites, we see the general characteristics of an extratropical cyclone. The highest wind speeds are found on the southeastern flank of the EXC100, which is most likely associated with the presence of the warm sector of the cyclone. On the contrary, lower wind speeds are found on the northern flank of the EXC100. The most intense precipitation is present just east
of the cyclone core. However, typical structures associated with extratropical cyclones, such as fronts, are hard to identify due to the low horizontal resolution of the model. When comparing the composites between the regions, some notable differences are found. Generally, EXC100s in the central Mediterranean are stronger in terms of wind speed than EXC100s in the eastern Mediterranean. Wind speed EXC100s (Fig. 8b) in the central Mediterranean exceed 25 ms$^{-1}$, whereas they barely exceed 20 ms$^{-1}$ in the eastern Mediterranean. EXC100s are statistically significantly stronger in the central Mediterranean for the
southeastern flank of the cyclone (warm sector). However, wind speed EXC100s in the eastern Mediterranean are significantly stronger on the northern flank.

It must be noted that the region defined as the central Mediterranean is located more north and, therefore, may be more heavily influenced by the stronger midlatitude storm tracks. Secondly, wind speed EXC100s and compounding events tend to have a lower core pressure than precipitation EXC100s in the order of 5 hPa. This holds true for both the EXC100s in the
central and eastern Mediterranean. Furthermore, precipitation EXC100s have a lower wind speed than wind speed EXC100s and vice versa, which is expected. Compounding EXC100s show a similar pattern compared to wind speed EXCs for both regions.

As expected, precipitation EXC100s have much higher precipitation rates than wind speed EXCs with differences of up to 10 mm. Compounding EXCs tend to produce precipitation rates comparable to those of EXCs with respect to precipitation.
There are statistically significant differences for precipitation, but compared to wind speed, there does not seem to be a part of the cyclone that is particularly affected by regional differences, and regional differences seem small in general.

The EXC100 composites for JJA, show much lower extreme values with respect to WS850 and 6-hourly precipitation (Fig. S3). The differences for WS850 and 6-hourly precipitation are in the order of 50% or more. This is most likely due to the fact that cyclones in the Mediterranean are less extreme in summer compared to winter. However, EXC100s in the
central Mediterranean tend to produce stronger wind speeds and precipitation rates than in the eastern Mediterranean, where the cyclone barely appears in the composites in either sea level pressure, precipitation or wind speed.

To characterise the life cycle of extreme cyclones, we show the temporal evolution of EXC10, EXC100, and EXC1000 in the central and eastern Mediterranean (Fig. 8). We do this, by selecting the most extreme value from each composite for every time step. Wind speed and precipitation increase and peak at **t0** before they decline again when the cyclone dissipates. Hardly any
significant differences between the central and eastern Mediterranean can be seen for the EXC100 for either precipitation, wind speed, or compounding events. However, the differences between the EXC1000 are significant most of the time, indicating a structural difference between less extreme cyclones in the two regions. In Fig. 8a, EXC10 stands out in terms of extremeness in precipitation in the central Mediterranean, as it reaches a maximum of around 14 mm/6h compared to the 8 to 10 mm/6h for the other EXCs, although the difference is only statistically significant compared to the EXC10 in the eastern Mediterranean at



t0 till 18 hours after t0. For the wind speed EXCs and compounding events (Fig. 8b and c), precipitation tends to peak before t0. The reason why the peak for precipitation in precipitation EXCs is not consistent with the other EXC types is most likely due to the method we applied: because we selected the EXCs with the highest precipitation at t0, then the peak is also most likely occurring at t0. Another notable difference between the eastern and central Mediterranean is that precipitation during the hours before t0 is generally higher in the central Mediterranean compared to the eastern Mediterranean (significant for wind speed and compounding EXC100s and EXC1000s in Fig. 8b and c), with differences of a few mm. However, the difference after t0 dissipates.





**Figure 8.** The evolution of three extreme cyclone statistics in the central Mediterranean (in black) and the eastern Mediterranean (in red). The values indicate the most extreme value of every composite for 6-hourly precipitation in mm (a–c), 850 hPa wind speed in ms$^{-1}$ (d–f), and sea level pressure in hPa (g–i) for precipitation EXCs (left column), wind speed EXCs (middle column) and compounding EXCs (right column). The results are shown from 30 hours before the extreme cyclone's minimum SLP until 30 hours after the extreme cyclone's minimum SLP. Shown are the values for EXC10 (dashed line), EXC100 (solid line) and EXC1000 (dotted line). The markers indicate whether the difference between EXC10 (square), EXC100 (circle), and EXC1000 (triangle), respectively, are statistically significant between the regions for a point in time (p=0.05).



When looking at WS850 (Fig. 8d–f), the higher wind speeds in the central Mediterranean from Fig. 7 are apparent throughout the entire cyclone life span. Contrary to the precipitation in Fig. 8a–c, these differences also hold after t0. The only exception is the wind speed for precipitation EXC100s (Fig. 8d), where there is hardly any significant difference until t0. However,

precipitation EXCs in the central Mediterranean seem to sustain higher wind speeds after t0. Something similar is found for the compounding events (Fig. 8f). Though it must be noted that most of the differences between the central and eastern Mediterranean are only significant for EXC1000s.

EXCs deepen quickly in terms of SLP before t0, whereas they fill up more slowly after t0 (Fig. 8g–i). SLP is generally lower in the central Mediterranean than in the eastern Mediterranean due to the more southerly occurence of cylcones in the

eastern Mediterranean. Interestingly enough, differences between EXC10, EXC100, and EXC1000 are larger for compounding events than precipitation or wind speed EXCs, especially in the central Mediterranean.

Considering the temporal evolution during summer (Fig. S4), the differences between the eastern Mediterranean and the central Mediterranean are small for precipitation EXC100s. The precipitation EXC100s in the central Mediterranean produce slightly more precipitation over the cyclone's lifetime Fig. S4a–c. However, the differences between the two regions for

wind speed EXCs, and compounding events are large. EXC10s and EXC100s produce much higher wind speeds for wind speed EXCs and to a lesser extent compounding events in the central Mediterranean compared to the eastern Mediterranean (Fig. S4d–f). Still, apart from wind speed EXC10s and EXC100s in the central Mediterranean (Fig. S4d), WS850 and 6-hourly precipitation is in the order of 50% less over the whole cyclone lifetime in JJA compared to DJF. Interestingly, neither in the central Mediterranean nor in the eastern Mediterranean do SLPs fall during the deepening phase, and neither do they rise again

after t0 (Fig. S4g–i). Wind and compounding EXCs in JJA seem to be only able to produce wind speeds higher than 15 ms$^{-1}$ in the central Mediterranean. It must be noted that our model is far too coarse to properly capture convective processes, which play an important role in summer, possibly leading to a misrepresentation of cyclone-related precipitation in JJA.

To identify what drives the differences between the regions and the EXC types, we show the EXC100 composites of the mean state in the upper atmosphere. In Fig. 9a–f, cold air masses are located northwest of the EXC100 centre and warm air

masses are located southeast of the EXC100 centre, highlighting the warm and cold sectors of the cyclones. The highest WS300 values are found south of the EXC100 centre, indicating that the jet stream is usually located south of the EXC100 centre.

The differences between the central and eastern Mediterranean in Fig. 9 are large. The jet stream south of the EXC100 centre in the eastern Mediterranean (up to 50 ms$^{-1}$) is much stronger than over the central Mediterranean (up to 40 ms$^{-1}$). Also, the jet stream remains present southeast of the EXC100 centre in the eastern Mediterranean, whereas the jet stream in the

central Mediterranean is only present south and southwest of the EXC100 centre. Given the location of our areas of interest, EXC100s in the eastern Mediterranean are generally located more south and closer to the subtropical jet. This explains the higher WS300 values in the eastern Mediterranean. Another difference between the two regions is that EXC100s in the eastern Mediterranean are related to much stronger intrusions of cold air in the lower troposphere northwest of the EXC100 centre (more than -8 °C compared to -4 ° C in the central Mediterranean). Although the negative T850 anomalies are stronger over the

eastern Mediterranean, EXC100s over the central Mediterranean are accompanied by stronger positive T850 anomalies over the southeastern flank of the EXC100 centre (more than 4 °C compared to 2 ° C in the eastern Mediterranean).





**Figure 9.** Same as Fig. 7 but now for 300 hPa wind speed in ms[-1] (green shading), and 850 hPa temperature anomalies in °C (red and blue contour lines) in panels a–f. In panels g–l, red-blue shadings show the 500 hPa geopotential height anomalies in gpm and the dashed white contour lines show the Rossby wave packet amplitude in ms[-1]. The grey contour lines indicate the SLP in all panels.

In Fig. 9g–l, the cyclone centres are often located just east of the lowest Z500 anomalies and the maximum RWP amplitude, indicating a westward tilting of the cyclone in the upper atmosphere. Also, EXC100 composites are accompanied by weak positive Z500 anomalies over the southeast of the EXC100 centre.

Wind speed EXC100s are often associated with higher WS300 values and T850 gradients. This again indicates that precipitation EXC100s require less dynamical forcing than wind speed EXC100s. However, the differences are larger for the subtypes within the eastern Mediterranean than in the central Mediterranean. Interestingly, positive T850 anomalies within the warm




sector do not differ between EXC100 subtypes within the regions, whereas negative T850 anomalies within the cold sector are usually stronger for wind speed EXC100s and compounding EXC100s than for precipitation EXC100s.

EXC100s in the central Mediterranean are accompanied by an areawise more negative Z500 anomaly than in the eastern Mediterranean. Precipitation EXC100s are associated with less strong Z500 anomalies than the other two EXC100 types (only going up to 200 gpm). The Z500 anomalies for precipitation EXCs in the eastern Mediterranean are especially small in size.

Negative Z500 anomalies are often associated with troughs, which can be linked to intruding Rossby waves, explaining why an RWP is present for all EXC100 types and regions. The centre of the RWP is located slightly west of the centre of the
Z500 anomaly. RWPs for EXC100s in the eastern Mediterranean have a larger amplitude (up to 38 ms$^{-1}$) than EXC100s in the central Mediterranean (up to 30 ms$^{-1}$). This indicates that intruding Rossby waves play a crucial role in cyclone development in the eastern Mediterranean. Strikingly, the RWP amplitude between EXC100 types for the same regions seems to be similar. It seems that compounding EXCs in the eastern Mediterranean are associated with a slightly stronger RWP amplitude than precipitation and wind speed EXCs. However, this difference is small and cannot be seen in the central Mediterranean.

The EXC100 composites for the upper-atmosphere during JJA (Fig. S5) show generally less intense values for WS300, T850 anomalies, Z500 anomalies, and RWP amplitude compared to the EXC100 composites in Fig. 9. The only exceptions here are the T850 anomalies for the central Mediterranean that are stronger in JJA by about 2 °C (Fig. S5a–c). Also, the relative differences between JJA and DJF are much larger in the eastern Mediterranean compared to the central Mediterranean. The difference between the two seasons indicates that the dynamical drivers of EXCs are less pronounced in summer, leading
to fewer extreme cyclones. However, in JJA, the WS300 and the RWP amplitude composites are strongest in the central Mediterranean, in contrast to DJF where WS300 and RWP amplitude are strongest in the eastern Mediterranean, suggesting the subtropical jet plays a much smaller role in the eastern Mediterranean in summer. The jet stream only plays a role for EXC100s with respect to wind speed in the eastern Mediterranean (Fig. S5e).

In Fig. 10 we show the time series of the maxima in WS300 (top row) and RWP amplitude (bottom row) during the cyclone
lifetime. As already shown in Fig. 9, we can see that at t0 the values for RWP amplitude and WS300 are higher in the eastern Mediterranean than in the central Mediterranean. Generally speaking, this holds true for most of the EXC life cycle. However, there are some exceptions to this. For the EXC100, the differences are only statistically significant for the wind speed EXCs (Fig. 10b). Also, after t0, WS300 increases for all EXC event types in the eastern Mediterranean, leading to a bigger difference for WS300 between the eastern and central Mediterranean. Also, there is only a small difference between WS300 values for
EXC10, EXC100, or EXC1000 during the life cycle of a cyclone in the eastern Mediterranean (compound events being the only exception). In the central Mediterranean, only the EXC10 stand out in terms of WS300 values. This would imply, that the strength of the jet stream only plays an important role for the most extreme cyclones.





**Figure 10.** Same as Fig. 8 but now for 300 hPa wind speed in ms$^{-1}$ (a–c), and Rossby wave packet amplitude ms$^{-1}$ (d–f).

Throughout the entire EXC life cycle, the RWP amplitude is also higher in the eastern Mediterranean compared to the central Mediterranean. This difference is significant for wind speed and compounding EXC100s. Strikingly, during the development

phase of the EXC, the RWP amplitude tends to increase in the eastern Mediterranean and decrease in the central Mediterranean. After **t0**, this gap diminishes, indicating that peak RWP amplitude plays a bigger role in EXC development for the eastern Mediterranean compared to the central Mediterranean.

The temporal evolution of WS300 and RWP amplitude during JJA is similar to the temporal evolution in Fig. 10. WS300 values stay more or less constant during the whole cyclone lifetime, whereas maximum RWP amplitude decreases after **t0**.

The main differences compared to DJF are that maximum values for WS300 and RWP amplitude are lower and that the regions with the highest values are the central Mediterranean compared to the eastern Mediterranean in DJF.





## 4 Discussion and conclusion

In this study, the impact of climate variability on Mediterranean cyclones is analysed for the period from 1500 BCE to 1850 CE using CESM. Moreover, we present the major characteristics of extreme cyclones in the region for this time period. While substantial research has explored how present and future climatic states affect the occurrence and intensity of Mediterranean cyclones (Raible et al., 2010; Lionello et al., 2016; Hochman et al., 2020), this study is novel in providing a baseline for Mediterranean cyclones within a multi-millennium perspective not affected by anthropogenic global warming.

We find that periods with higher and lower cyclone frequency and intensity in the Mediterranean exhibit a clear multi-decadal variability in the order of 5% from the multi-millennial mean. The phase of the most dominant atmospheric modes of circulation only explains little of the variance in cyclone frequency and nothing of the cyclone-related wind speed or precipitation in the Mediterranean, despite them having a strong influence on cyclone-related features in the North Atlantic and northern Europe. Volcanic eruptions can cause a significant increase in cyclone frequency in the eastern Mediterranean, but for the Mediterranean as a whole, the signal is rather weak. Generally, we find significant impacts of volcanic eruptions over large parts of North America, the Atlantic Ocean, and its neighbouring seas.

Raible et al. (2018) used an older version of CESM but with a higher horizontal resolution ($1.0° \times 1.0°$) to track cyclones in the North Atlantic and over Europe. The biases they observed are similar to the results presented here (Fig. 2), with an overestimation of cyclone frequency near the Greenland ice sheet and CESM being too zonal. We have shown that compared to ERA5 reanalysis our model simulation and the cyclone detection and tracking algorithm captures the extreme cyclones in the Mediterranean Sea well, even though the horizontal resolution of the climate model used is crucial (Flaounas et al., 2013). The weak relations between the cyclone frequency and the most dominant modes of circulation in the Mediterranean (Fig. 6) are similar to those found by Seierstad et al. (2007) and Walz et al. (2018). However, the low resolution of our model may be an additional reason why we find no relation between the most dominant modes of circulation and the cyclone-related precipitation or wind speed. It is reported that in CMIP5 models with relatively coarse horizontal resolutions, the winter North Atlantic storm tracks tend to show too zonal structure or with a southward displacement (Zappa et al., 2014; Müller et al., 2018). These biases also influence the NAO variability, which is related to the extratropical and Mediterranean cyclone variability. Although the CESM version with its coarse resolution used in our study contains the same problem, it still provides a long transient simulation of the last 3350 years, therefore, allowing investigating long-term temporal variability of cyclones and related features.

We also find that the variability of Mediterranean cyclones (Fig. 4) is very similar to what Raible et al. (2018) observed for North Atlantic cyclones in the aforementioned earlier version of CESM. They revealed that the cyclone variability is not driven by solar irradiance variations. This confirms our result, which shows no significant relationship between the Mediterranean cyclone variability and solar irradiance. However, Raible et al. (2018) found no imprint of volcanic eruptions on cyclone frequency. This contradicts our result, which is more in line with Andreasen et al. (2024). Andreasen et al. (2024) found an increase in cyclones over the subtropics and high latitudes and a decrease over the mid-latitudes after volcanic eruptions due to larger meridional temperature gradients and a lower tropopause. Though it must be noted that the eruptions in Andreasen et al.



(2024) are much stronger than the eruptions in the simulation of Raible et al. (2018) who consider eruptions after 850 CE. This also explains why our result is more similar to Andreasen et al. (2024), as our simulation includes stronger eruptions due to the longer time span.

The composite analysis shows that wind speed EXCs are generally stronger in the central Mediterranean as compared to the eastern Mediterranean. The difference between the two regions for precipitation EXCs is much less clear, and overall, there does not seem to be a large difference between the two regions for precipitation EXCs. For wind speed and compounding EXCs we observe a precipitation peak that occurs 6 to 12 hours earlier than the wind speed peak which coincides with $t0$. This is consistent with Messmer and Simmonds (2021) and Raveh-Rubin and Wernli (2015), who found that precipitation peaks before wind speed in the ERA5 reanalysis, and with Booth et al. (2018), who showed that the precipitation on average peaks 12

hours prior to the dynamical strength maximum of a cyclone based on satellite data. Due to our selection criteria of selecting cyclones with the highest precipitation rates at $t0$, we do not observe this for precipitation EXCs. Our findings indicate that EXCs in the central Mediterranean can have a higher potential impact on society due to their higher potential wind speeds. This is amplified by the fact that EXCs in the central Mediterranean tend to be more extreme in terms of wind speed over their entire life cycle and not only at $t0$.

(Flaounas et al., 2015b) performed a similar composite analysis of intense cyclones using a regional model with a higher horizontal resolution (20 km horizontal resolution) but considering the Mediterranean basin as a whole. Despite the higher resolution, the structure and the location of the region with the most intense precipitation are similar to the results presented here, which gives us confidence in the ability of CESM to produce extreme cyclones despite the low resolution of our model. They also suggest that the subtropical jet might play a crucial role in cyclone development by providing barotropic shear.

Despite the fact that the subtropical jet is clearly more dominant in the eastern Mediterranean in winter (Fig. 9 and 8), this does not result in stronger cyclones over the region and may not be the driving factor for the development of EXCs. max (Homar et al., 2007) showed that the central Mediterranean experiences more intense Mediterranean cyclones than the eastern Mediterranean. This is in line with our results where we observe more intense wind speed EXCs in the central Mediterranean, since wind speed EXCs require a deeper and better developed cyclone than precipitation EXCs (Fig. 7). However, Flaounas

et al. (2023) showed that there is no difference between the two regions in the occurrence of intense cyclones using composite tracks in the ERA5-reanalysis. Nevertheless, our simulation has the advantage that, due to the long time period, the effect of natural variability should be averaged out, adding to the hypothesis of Homar et al. (2007).

To conclude, there is no obvious single driver of Mediterranean cyclone variability, Mediterranean cyclones vary on multi-decadal scales with an amplitude of up to 5% from the multi-millennial mean. Additionally, our findings indicate that cyclones

may have a stronger impact on nature and people in the central Mediterranean than in the eastern Mediterranean, especially in terms of wind speed extremes. Our study offers a climatological reference baseline for understanding extreme cyclones in the region. Having this reference baseline is also beneficial for quantifying the effects of future climate change on Mediterranean cyclones. Nonetheless, the low resolution of the model simulation used is a major drawback. Thus, future work on the long-term variability of cyclones using a higher-resolution model is needed, as pointed out by Flaounas et al. (2013). For instance, a



regional climate model can be used to downscale parts of this CESM simulation to provide more realistic insights into extreme Mediterranean cyclones and their intensification processes.

*Code availability.* The cyclone tracking was performed with the detection and tracking scheme of Blender et al. (1997) and is available on request. The other analysis steps were performed with python scripts. As they are standard methods, they are not uploaded to a repository. These scripts are available on request.

*Data availability.* The ERA5 data are provided by Copernicus Climate Change Service Climate Data Store (CDS) from their website at https://cds.climate.copernicus.eu (last access: 9 Nov 2023). Post-processed CESM and ERA5 data used for the study are available at https://doi.org/10.5281/zenodo.13619444 Complete CESM1.2.2 data are locally stored and are available upon request

*Author contributions.* OD, MM, WMK, and CCR contributed to the design of the study. WMK carried out the climate simulations. OD performed the principal analysis and wrote the manuscript under the supervision of CCR. MM, WMK, and CCR provided critical feedback on the results and drafted the manuscript together with OD. All authors contributed to the writing and scientific discussion.

*Competing interests.* The authors declare no conflict of interest.

*Acknowledgements.* We acknowledge the Swiss National Supercomputing Centre (CSCS) in Lugano, Switzerland, for providing the necessary computational resources and supercomputing architecture to perform the simulations under project number s1248. OD and CCR received funding from the Swiss National Science Foundation (grant nos. 200020_172745 and IZCOZ0_205416).



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
