# Peer review of "Characterization of the mean and extreme Mediterranean cyclones and their variability during the period 1500 BCE to 1850 CE"

_EGUsphere, 2024_

## Author Comment (AC1)

**We thank the reviewer for their constructive and helpful comments, which will provide a more robust approach and increase the readability of our manuscript.**

**Major comments**

1: *Given a baroclinic growing cyclone, I would expect maximum precipitation (rates) on average around the time of maximum intensification (e.g. Papritz et al., 2021). This is also reflected in the precipitation composites in e.g. Figure 8, where the maximum precipitation for the extreme wind composites (Fig. b and c) is before the minimum pressure is reached. That the precipitation maximum is still at the minimum pressure for the extreme precipitation composites, is probably due to the sampling on the maximum precipitation, as the authors also remark themselves. That the cyclones grow on average baroclinically is visible in some of the composites a westward tilt of the minimum pressure with height. Can the authors argue why they did made this choice?*

We agree with this comment that it is well-known that cyclone-related precipitation peaks before the cyclone reaches its mature stage (i.e. when the cyclone reaches its lower core pressure) (Booth et al., 2018, Papritz et al., 2021). We decided to look at precipitation at the time of lowest core pressure to let it be consistent with wind speed. However, we do agree that this is not the best way to select extreme precipitation cyclones. Therefore, we will repeat the analysis for extreme precipitation cyclones by selecting those cyclones which have the highest precipitation rate independent of the core pressure. Still, the cyclone state with the highest precipitation rate must be located within one of the corresponding boxes of Fig. 1, defining the central and eastern Mediterranean. Our expectation is that, for example, Fig. 8 a,d,g, and Fig. 9 a,d will change and show, e.g. that the highest precipitation rate is reached before the minimum core pressure.

2: *The minimum pressure is not the best way to assess cyclone strength, since it depends on both the latitude (since there is a equator to pole gradient of pressure) and the size, since a minimum pressure does not necessary indicate a strong gradient of pressure. Can the authors argue why this would be a sensible choice, and have they tested for example using the gradient of gpm per 1000 km, since given the cyclone detection algorithm they have this information available?*

We fully agree with the reviewer and can see that the reviewer might have misinterpreted our results. We use wind speed as an indication of strength. We only use minimum core SLP to illustrate the life cycle and the mature stage of a cyclone. In Fig 8 d,h, we show the minimum core SLP to give the community the opportunity to compare our results with existing literature, as still most of the studies use minimum core SLP as a strength measure. Note that the strength of the cyclone in Fig 8  is based on its related cyclone-related precipitation and wind speed at the time of minimal core pressure. As shown by Pfahl and Sprenger (2016), this works well for wind speed. For precipitation, this method has drawbacks, as explained in the reply above.

As a side remark, the issue of minimum core SLP versus gradient might not be so relevant here in this study, as our region does not span over a large range of latitudes. We show the gradient pressure relationship to illustrate this here in the response (see below Fig. S1).
To adapt this point we will revise the text accordingly.

[Figure]

*Fig S1: Core pressure (blue) versus gradient (red) for the 10 (striped lines), 100 (normal lines) and 1000 (dotted lines) most extreme cyclones in the central Mediterranean (left) and the eastern Mediterranean (right).*

In Fig. S1, we show the relation between gradient and core pressure for EXC10, EXC100 and EXC1000, similar to Fig. 8g–i in the manuscript. The relation that the highest gradient is achieved when the core cyclone pressure is the lowest is evident for the central Mediterranean and is also true most of the time in the eastern Mediterranean. We assume this will be the same for other measures like cyclone depth. Hence for individual tracks, using the time of minimum core pressure is a good indication of cyclone maturity.

*3: In several parts of the paper (e.g. line 50) the authors argue that the modes of variability influence the occurrence and strength of Mediterranean cyclones. This would suggest some causal link. However, I would argue that it is merely a correlation, as the authors also write for e.g. the link to the NAO (line 48). I would therefore suggest rewriting the text at these points a bit.*

This is a fair point. We agree that our wording is too strong and the general atmospheric circulation modes are not a cause of cyclones but merely a correlation. These points will be addressed in the revised version.

**Minor comments**

*Line 8: Does this variability refer to the variability of cyclone frequency?*

We agree that this was not fully clear. We refer to a set of cyclone characteristics, not just frequency. The variability of cyclone frequency, 90th percentile cyclone-related precipitation and 90th percentile cyclone-related wind speed are in the order of 5%.

To clarify this, we will change it to:

"We found that Mediterranean cyclone characteristics exhibit pronounced multi-decadal variability in the order of 5% throughout the entire late Holocene with respect to several cyclone-related properties."

*Lines 47-49: This is an almost exact repetition of the sentence before, so I would suggest rewriting it and remove one the lines. I think this sentence is clearer than the previous, so I would suggest keeping this sentence.*

We agree, we would rather keep the latter sentence.

Yes, "they" refers to proxies. This sentence should be rewritten anyway to make it flow better, for example:

"Yet, since proxies are usually only sensitive to temperature and precipitation, and not to wind and pressure, it is complicated to reconstruct cyclone activity directly (Raible et al., 2021)."

*Line 92: The focus on extremes could be motivated a bit more since it is a vital part of the manuscript.*

Agreed, some more background literature will be provided.

*Line 103: 1.9 x 2.5 degree longitude and latitude respectively?*

Yes, indeed, this will be clarified in the text.

*Line 119: Given that you use the Z1000, you probably detect minima in the geopotential (height) and not pressure directly?*

Indeed, the sentence should be changed a bit to make it more consistent, for example:

"The algorithm identified local minima in the Z1000 field"

It should be the mean gradient averaged over 1000 km. This will be changed to:

"A minimum mean gradient of at least 20 geopotential meters (gpm) per 1000 km is achieved"

*Line 124-125: I do not understand it completely, since as far as I understand the authors track the cyclones on a 6-hourly resolution, so why is this criterion applied daily?*

It means the new cyclone centre cannot be more than 1000 km away from the previous centre if the time difference were to be 24h. However, since we have 6-hourly data, this basically means that the new cyclone centre cannot be further away than 250 km. We agree this is not very clear, though, and this will be clarified in the text as follows:

"The new minimum of the cyclone must be within 250 km of the previous cyclone minimum."

*Line 143: Are these eruptions used, independently, where they occurred on earth?*

Yes, they can occur anywhere on Earth. Most of the major eruptions have a tropical origin and thus can affect both hemispheres. This will also be emphasized in section 2.3.

*Line 144: Why is there a time frame of 5 years used before an eruption and only a time frame of 2 years after the eruption?*

The time frame of 5 years before the eruptions will provide enough years to characterize internal climate variability (note that we do this for 20 eruptions, which equals 100 years of "undistvrbed climate variability"). We only use the first two years after an eruption as these are the years when the impact of the forcing is strongest, so we expect to see the strongest signals of volcanoes. Our results already indicate that the impact is not strong, so extending it to longer periods after an eruption will decrease the signal-to-noise ratio.

*Line 154: This region could be indicated in one of the Figures.*

Since the region we chose to define the PC-based NAO is a standard region as defined by Hurrell et al. (1995), and since we don't really see a suitable figure to include a region this large, we think our paper will remain more structured if we just leave the boundaries of the region in the text as defined in line 154.

*Line 167: See above major remark, why not selecting on the time step of maximum intensification, since this is one would expect strongest precipitation rates?*

See response to major comment 1.

*Line 176: How many cyclones are detected in total, or in other words, what is the fraction of selected 'extreme' cyclones?*

That's a fair point, and we will include the numbers in the text.

*Lines 200-205: The authors argue that the storm tracks are too zonal compared to ERA5 in the CESM model. However, if I would for example look at the DJF climatology, I would almost argue the opposite: there are relatively more cyclones detected at the northern side of the storm tracks, and less at the southern side. Can the authors explain why they argue that the storm tracks are too zonal in CESM?*

The comment that CESM is too zonal mainly refers to the Atlantic, where, in our opinion, the lack of cyclones penetrating the subtropics is evident, and we allocate this to a zonal bias. However, as the other reviewer pointed out, this is probably also a consequence of a northern shift in the storm tracks. This will be accounted for in the corrected version.

*Line 210: See previous remark, isn't it more a northward shift of the storm tracks (away from the Mediterranean)?*

See previous comment for line 200-205.

*Line 229: I might have missed this, but the 850 hPa temperature related to the cyclones is calculated in a certain area/radius around the cyclones? And the plotted temperature in Figure 4 is then the average over all cyclones occurring in a certain year?*

Fig 4a refers to the 30-year running mean T850 anomalies for the two boxes in Fig 1b combined. Fig 4a is not related to any cyclone-related metrics and just acts as a proxy for the state of the climate of the last 3500 years. We will write this more explicitly in the updated version of the manuscript.

*Line 259: See remark above, I would be careful to describe these modes as 'drivers of the circulation'.*

See reaction to major comment 3.

*Line 268: See previous remark*

See reaction to major comment 3.

*Line 287-288: I think this sentence could be moved to the methodology section, since it is not related to results shown.*

Given that we find no relevant correlation at all and the fact that this sentence does not fit into the rest of the narrative, we have dedicated to remove this sentence from the manuscript.

*Line 300: To what does the three different EXC types refer too? I assume it is the extreme precipitation, wind and compound composites? This could be further clarified.*

Although they are mentioned in section 2.5, we agree it would be useful to note them down here again for the sake of clarity. This will be included in the revised version.

*Lines 320-321: I do not understand what the authors try to argue here, can the authors clarify their argument here?*

When observing the wind speed composites Fig. 7, visually, there are wind speed and compounding EXCs that are much stronger in the central Mediterranean compared to the eastern Mediterranean (Fig. 7b vs 7e and Fig. 7c vs 7f). This is also evident from the dots indicating statistical differences between the two. The argument here is that grid cells with statistical significance have a much more coherent pattern for the wind speed composites (which mostly coincide with the areas with the highest wind speeds) than the precipitation

composites. The statistical significance pattern for the precipitation compounds is much less coherent. We understand, however, where the confusion comes from and this part should be clarified in the text.

*Lines 337-338: Given this possible preselection bias, I would strongly suggest the authors to look at the sensitivity of this choice.*

See reaction to major comment 1.

*Line 352: Is Figure 8 then DJF?*

Yes, this obviously should be highlighted.

*Line 369: The authors write that the jet stream remains at certain position, which suggests that the jet stream remains at the certain position over a time period, but I don't think that is what the authors mean.*

This is indeed not what we mean, and we appreciate the reviewer's sharp observation. What is meant here is that the jet stream is located southeast of the cyclone during the mature phase. Hence, we will change the sentence to the following:

"Also, the jet stream is located southeast of the EXC100 centre in the eastern Mediterranean, whereas the jet stream in the central Mediterranean is only located south and southwest of the EXC100 centre."

*Line 378: This already suggests that the (detected) cyclones grow baroclinically*

Thank you for the suggestion. We will include the sentence.

*Line 385: What is meant with an areawise more negative anomaly?*

This simply means that EXCs in the central Mediterranean are accompanied by a larger negative Z500 anomaly with respect to size (i.e. a larger trough in size). We will clarify this.

*Lines 459-465: I would suggest writing the abbreviation EXC in full, since this probably would clarify the text.*

We agree this will be changed.

*Caption Figure 4c: I think it is a precipitation rate (in mm/6h), as also described in the label of the y-axis?*

Yes, this is an error, and will be corrected.

*Figure 7 and elsewhere: I would suggest to make the text of the regions Central and Eastern Mediterranean bold, the first time I read the figure labels I was confused because I read them as 'central Mediterranean longitude'*

We thank the reviewer for the suggestion, and we agree that the suggested correction would make the labels less confusing.

**References**

Booth, J. F., Naud, C. M., & Jeyaratnam, J. (2018). Extratropical Cyclone Precipitation Life Cycles: A Satellite‑Based Analysis. In Geophysical Research Letters (Vol. 45, Issue 16, pp. 8647–8654). American Geophysical Union (AGU). https://doi.org/10.1029/2018gl078977

Hurrell, J. W. (1995). Decadal trends in the North Atlantic oscillation: regional temperatures and precipitation. *Science*, *269*(5224), 676–679. https://doi.org/10.1126/science.269.5224.676

Papritz, L., F. Aemisegger, and H. Wernli, 2021: Sources and Transport Pathways of Precipitating Waters in Cold-Season Deep North Atlantic Cyclones. *J. Atmos. Sci.*, **78**, 3349–3368, https://doi.org/10.1175/JAS-D-21-0105.1.

Pfahl, S., & Sprenger, M. (2016). On the relationship between extratropical cyclone precipitation and intensity. In Geophysical Research Letters (Vol. 43, Issue 4, pp. 1752–1758). American Geophysical Union (AGU). https://doi.org/10.1002/2016gl068018

---

## Author Comment (AC2)

**We thank the reviewer for their constructive and helpful comments, which will provide a more robust approach and increase the readability of our manuscript.**

*1: The structure of the results section is imbalanced. On the one hand, the first half presents an overview of different aspects, where many results apply to the North Atlantic rather than the Mediterranean (Figs. 2, 5, 6). They may be relevant for the Mediterranean but this is not discussed. For instance, not much impact of volcanic eruptions is found, nor much link with atmospheric modes. On the other hand, the second half describes intense cases only. It is very detailed but contained in a single section. Some reorganization is required here, with important results highlighted and others streamlined.*

Thanks for the reviewer's suggestion. However, we do not fully agree with this comment. One of our intentions of including the Atlantic is because the region is upstream to the Mediterranean. In addition, we would like to argue that the Atlantic offers a possibility of a comparison of the role of volcanic eruptions and atmospheric modes of circulation in the Mediterranean.

Responding to the reviewer's comment, we can see that the arguments we attempted to deliver were not clear. So we will adjust the text to motivate the sections better and explain the relevance of the Mediterranean better than in the current form.

We also agree that some sections need a more smooth transition and some figures show too much information. For Figs 2 and 3, we will only show the results for DJF and JJA, as MAM and SON are not discussed in other sections of the paper, and move the results for MAM and SON to the supplementary section. Furthermore, we also suggest removing the middle and right columns of Fig. 6, as the discussed modes of circulation have no significant impact on cyclone-related wind speed and precipitation in the Mediterranean. These changes will help to move the focus on the Mediterranean.

*2: Why are results compared between the western/central and eastern Mediterranean? Dynamical differences between these regions are not introduced, despite the large body of literature about Mediterranean cyclones. In contrast, Genoa lows, Vb cyclones, Sharav cyclones and Medicanes are introduced but not further discussed. Thus, the reader does not know what to learn from the comparison.*

We thank the reviewer for raising this point. We do not justify enough why we differentiate between the regions. Partly, this is due to the differences we found in the results, but we agree that we have to justify this more in the introduction. Therefore, we will include a more solid basis (by updating the literature) for the differences between the regions with respect to cyclones in the introduction.

3: *Why is CESM compared with ERA5 for the period 1980–2010 only and not for the full period 1940–present?*

The 30-year reference periods are used to define climate averages as defined by the WMO. Also, the time period suggested by the reviewer overlaps with a strong increase in global temperature and its potential effects on global circulation. Additionally, the pre-1979 ERA5 satellite observations were sparse, which can cause the risk of including biased data with respect to the storm tracks. Note also that extending the period to the present would mean

that we compare observations with emission scenario-driven climate (CESM). Therefore, we see no reason to change this time period.

4. Methods are not always clear and definitions are sometimes repeated, which blurs the interpretation of results.

We agree that some definitions in the methods section are defined vaguely or repeated unnecessarily. With the reviewer's  suggestions in the minor comments below, we will address them and make sure the methods section overall becomes clearer and easier to understand.

5. In the results section, please systematically refer to the figure and panel that is discussed. Currently, one has to guess where to look at. Also, figures tend to contain many panels, which are not all discussed and could likely be removed to focus on the main results.

We agree that a lot of panels are not properly referred to, especially in the last few figures - we thank the reviewer for pointing them out. Additionally, in the reply to major comment 1, we suggested changes to some figures to make the information conveyed in the paper as a whole more concise.

**Minor comments**

*l. 3 "their variability in the late Holocene is poorly understood": more precisely?*

More precisely, their spatial and temporal variability in the late Holocene. This will be changed to:

"their spatial and temporal variability is poorly understood"

*l. 8 5% in what?*

The variability of cyclone frequency, 90th percentile cyclone-related precipitation and 90th percentile cyclone-related wind speed are in the order of 5%.

This will be changed to:

"We found that Mediterranean cyclones exhibit pronounced multi-decadal variability in the order of 5% throughout the entire late Holocene with respect to several cyclone-related properties."

*l. 9 the relation is described as "weak" in the conclusions*

We will add "weak relation" in the abstract to clarify.

*l. 24–25 What kind of variability and connection? This is the main motivation for the paper, thus requires (way) more details. Perhaps it is discussed below but it is unclear at that point.*

We agree this sentence is unclear. We propose rewriting it in the following way:

"However, the factors driving the variability of Mediterranean cyclone characteristics, especially extreme cyclones, are not fully understood."

*l. 40–41 References are expected here*

The paper by Cavicchia et al. (2013) will be included to refer to the Medicane climatology, and the sentence will be changed to the following (including that the peak also takes place in winter:

" a special type of Mediterranean cyclone is the so-called Medicane, which is a hybrid system between tropical and extratropical storms and often occurs in autumn and winter (Cavicchia et al., 2013)"

*l. 42 I don't fully agree: see, e.g., the devastating cyclone Daniel of September 2023*

This is a fair point, but we argue that extratropical cyclones are much more likely to occur in the Mediterranean, and their cumulative impact is much larger. We will weaken the statement to:

"However, due to their rarity, their overall socio-economic impact is not as large as that of extratropical Mediterranean cyclones"

*l. 43–44 Feser et al discuss North Atlantic cyclones whereas Flaounas et al discuss Mediterranean cyclones*

The reference of Feser et al. (2015) is with respect to the decadal variability this study observed in extratropical cyclones in NW Europe and the Atlantic. The reference of Flaounas et al. (2022) is with respect to section 2.2, where the relation between Mediterranean cyclones and teleconnection patterns is discussed. However, discussing the relation for extratropical cyclones in a more general sense may make more sense. Therefore, instead of Flaounas et al. (2022), we will include the references of Seierstad et al. (2007) and Walz et al. (2018) in the text.

*l. 46 Wintertime precipitation correlates negatively with NAO but positively with cyclone frequency*

This is an error in the structure of the sentence. The sentence should be:

"The NAO correlates negatively with the frequency of cyclones in the Mediterranean (Raible et al., 2007) and also negatively with  wintertime precipitation over the Mediterranean (Brandimarte et al., 2011; Montaldo and Sarigu, 2017).

This should be of a smaller spatial scale:

" Mediterranean cyclones, which are usually of a smaller spatial scale than other extratropical cyclones, suffer even more from the low resolution in GCMs (Flaounas et al., 2013)"

With low-frequency decadal to multi-decadal time scales are meant. Low frequency will be replaced in the text with the latter.

*l. 88 It would be worth citing and discussing the few studies, as the impact of volcanic eruptions on cyclones is investigated in the paper*

There are very few studies directly investigating the link between volcanic eruptions, and the ones we found are mentioned in the two sentences before. Though with some extra research, we did find an earlier study by Fischer-Bruns et al. (2005), who found no link between volcanic eruptions and storm activity. This study will be included in the introduction as well.

Brackets will be included (Kim et al., 2021).

See response to major comment.

*l. 125 I don't fully understand the condition "across" 24 hours*

It means the new cyclone centre cannot be more than 1000 km away from the previous centre if the time difference were to be 24h. However, since we have 6-hourly data, this basically means that the new cyclone centre cannot be further away than 250 km. This will be clarified in the text as follows:

"The new minimum of the cyclone must be within 250 km of the previous cyclone minimum."

*l. 134 missing "in" Raible et al*

Brackets will be included (Raible et al., 2018).

*l. 134–138 What is the meaning of fitting a Gaussian function to the geopotential field on a single grid point? Also, rather than describing first the "traditional" selection and then the adaptation, I recommend describing straight away what is actually used here.*

We agree that this part is a bit confusing, and we want to highlight that we do not fit a Gaussian function to one grid cell, as that would be impossible. What we meant is that sometimes the radii are so small that they do not extend over more than one grid cell. Still, to clarify, we decided to rewrite the last paragraphs of section 2.2 as follows (shortening it significantly):

"The cyclone identification and tracking method provided a variety of cyclone characteristics, such as the cyclone position, the radius of a cyclone, the cyclone depth, the core (central) pressure, and cyclone-related mean and extreme precipitation and wind speed. To compute the cyclone radius, first, a Gaussian function was fitted to the Z1000 field, assuming that the cyclone was azimuthally symmetric (Schneidereit et al., 2010). The cyclone radius is then defined as the distance between the cyclone centre and the point of 1.5 standard deviation (which represents the middle between the first and second inflection points), as done by Messmer and Simmonds (2021). The depth of the cyclone is defined as the difference between Z1000 in the centre of the cyclone and the Z1000 mean over the area of 1000 × 1000 km2. To calculate the cyclone-related wind speed and precipitation, the maximum value of wind speed and precipitation of all grid cells within a cyclone's radius were considered (Raible et al., 2018). We also applied the tracking algorithm to the ERA5 Z1000 field to evaluate the Mediterranean cyclone characteristics in CESM, mostly focusing on the cyclone tracks."

*l. 148 typo "significance"*

Will be corrected.

*l. 159 rendering issue*

Will be corrected.

*l. 161–165 not sure what "western Mediterranean" means here; actually the CMED region in Fig. 1 largely contains the region usually referred to as the West Med. Thus, for consistency with previous studies, and considering that only two regions are compared here, I recommend naming them West/East Med.*

This is a fair point, and we will change it in the manuscript.

*l. 166–177 The precipitation and wind speed metrics are repeated several times in the paragraph but in an inconsistent fashion (with/out max, with/out t0); I recommend defining them once for good and then simply using precipitation and wind speed (as in l. 177). Also, which radius is used along the track for precipitation and wind speed?*

According to the reviewer's comment, we updated the paragraph as follows:

"To do this, we ranked them based on the maximum 6-hourly cyclone-related precipitation (hereafter just precipitation) and 850 hPa wind speed (hereafter just wind speed) at the time when the cyclone reaches the lowest core sea level pressure within its track (hereafter t0)."

The rest of the paragraph will be corrected accordingly.

*l. 173 what is the untransformed distribution?*

For the selection of extreme precipitation cyclones, the assumption is that one picks the cyclones from the tail of a Gaussian distribution. This clearly is not the case for precipitation, and therefore, we compute the square root of the cyclone-related precipitation so the distribution looks more Gaussian.

To clarify this, we will explain this in more detail in the manuscript:

"The distribution for precipitation is skewed towards extreme values and values close to zero. Therefore, all 6-hourly precipitation rates below 1 mm were excluded, and we computed the square root of the remaining precipitation to assume a Gaussian distribution for precipitation."

*l. 177 here and elsewhere: CMED and EMED to keep the same terminology*

See comment for l. 161-165.

*l. 180 t0 already defined*

Agree and the sentence will be changed to the following:

"For each cyclone track, we set the reference time t0. Every time step of the track that occurred after t0 obtains a positive index, and the time steps of the track that occurred before t0 receive a negative index."

*l. 185 missing "at" t0?*

Indeed missing "at", will be included.

*l. 186 variations in what? I don't get the point here*

This refers to the "variations of the weights of the grid cell". Due to the small range of latitudes and the southerly location of the Mediterranean, we deem the differences in weights of the grid cells insignificant for the EXC composites. However, the sentence should be rephrased to make it clearer:

"Since all the cyclones were located in the Mediterranean basin t0 and, therefore, within a relatively small range of latitudes, variations due to differences in latitude weights of grid cells were considered insignificant and, therefore, negligible."

*l. 188 Why 30h? And missing "for" t0?*

30h is ratheran arbitrary value. We deem it enough to capture the intensification and decaying phase of the cyclone. We also tried plotting 48h before and after the cyclone, but it did not provide any extra insights. And indeed, "for" is missing.

*l. 191 are all fields averaged over the area of both regions for all cyclones?*

Bullet point number 4 should be clearer. We do not mean we compute averages of all fields that we have. We will compute spatial averages of the fields noted in the paragraph below. To clarify this, we will change bullet point 4 to the following:

"For every time step within these 30 hours before and after t0, we compute spatial averages for EXC10, EXC100, and EXC1000. This was done for both regions for cyclones associated with cyclone-related precipitation, wind, and compound extreme. Cyclone tracks, which did not appear at any of these time steps, were ignored for the temporal means."

*l. 192 Cyclone tracks "that" did not appear (and no comma)*

Will be changed.

*l. 195 Some details are expected about the RWP amplitude. And typo: Rossby wave "packet"*

Will be included, and the typo will be corrected.

*l. 200ff In the discussion on Fig. 2 please indicate which panel is referred to*

We will refer to the panels more consistently in this part of the text.

*l. 204 Fig. 2 suggests a northward shift rather than a zonal vs wavy storm track*

We agree that Fig. 2 also shows a northward shift of the jet. However, we would also argue that it is not just a northward shift of the jet, but also a decrease in waviness (e.g. CESM not capturing cyclone activity around the Azores in DJF). The sentence will be changed to:

 "Thus, the biases generally indicate that storm tracks in CESM are too zonal and shifted northward compared to the storm tracks in ERA5."

*l. 209–214 Any evidence for these assumptions? Otherwise they sound speculative*

In the introduction (line 31), we mention that about 20% of Mediterranean cyclones originate in the Atlantic. Now, although this does not account for the 50% underestimation we see in Fig. 2, we hypothesize that a less wavy and more northerly jet stream would contribute to fewer cyclones penetrating the Mediterranean and would also hamper conditions for lee cyclogenesis. We argue this is not merely speculation.

Will be changed.

To avoid speculation, we remove the sentence hypothesizing about the heat lows.

*l. 224 where is it most often underestimated by 50%? it is unclear where this number comes from*

This refers to the tails of the distribution of Fig. 6m–p, and the difference is about 50%. To clarify, we will rewrite it to the following:

"CESM struggles to reproduce high cyclone-related precipitation events in all four seasons, where the high-end tails of the distributions are most often underestimated by 50%"

*Fig. 3 wind speed is the max value here, while it is defined as the value at t0 in Section 2.5*

We appreciate the reviewer's sharpness here. We agree it is not very consistent to use max wind speed in a cyclone track instead of the wind speed at t0, so this will be adjusted in Fig. 3. However, we do emphasize that we do not expect a significant difference here.

*l. 234 the 30y running mean does not show similar patterns and is not appropriate for tendencies*

Our statement in line 234 is not worded properly and, therefore, causes confusion. What we mean is that in both Fig. 4a and Fig. 4b multidecadal variability is clearly visible, not that the variability in Fig. 4a and 4b are correlated. To clarify this, we change line 234 to the following:

"The cyclone frequency in the Mediterranean (Fig. 4b) also exhibits a clear multidecadal variability."

*l. 235 is this definition of cyclone frequency different from that used elsewhere in the paper?*

Slightly, as we sum the individual cyclone centres per time step in both regions for every month. However, Fig 4b does not take into account the cyclone radius as in Fig 1. To clarify the differences, we will change the label of Fig 4b to "total cyclone time steps".

This sentence can be shortened to:

"Only a small region with a significant increase in cyclone frequency is found in the eastern Mediterranean, but it still accounts for an increase of 30% in this region."

This should, of course, be important.

*l. 263–265 Please refer to the corresponding panels*

Correct references to panels will be included.

*l. 266 Fig. 6 is already presented before*

Good point. The first paragraph will be rewritten to make it flow better.

*l. 287 this sentence is surprising, as sea ice anomalies are not discussed in the methods*

After reviewer #1 also highlighted this, and since this sentence does not really fit in the rest of the manuscript, we have decided to remove it.

*l. 289 this should be mentioned earlier, and clarified that Fig. 6 shows DJF only*

That is a fair point, of course, DJF should be included in the first paragraph of section 3.4

*l. 303 the presence of the warm sector of the cyclone (I) could be verified and (II) does not dynamically explain the highest wind speeds (see, e.g., Raveh-Rubin and Wernli 2015, or papers for the North Atlantic)*

It is true that the presence of the warm sector is not the cause of the highest wind speeds. However, it is also obvious from Fig. 9 that at t0, the location of the warm sector and the highest wind speeds are correlated. To not cause any confusion or speculation though, we will discuss this in the paragraphs where Fig. 9 is discussed. Consequently, this discussion in this section of the paper will be removed.

*l. 309–311, 320--321 Please refer to the corresponding panels in Fig. 7*

Will be included.

*l. 312–313 not only northward but also (obviously) westward! Western/central and eastern Mediterranean cyclones have different dynamics, which should be discussed in the introduction (see, e.g., Doiteau et al. 2024, or older papers)*

That is a very fair point, and as discussed in major comment 2, we agree there should be more emphasis on the differences between the two regions. Hence, we will give a more solid literature basis in the introduction.

*l. 316 it is expected indeed; I don't quite get the point at showing precipitation for windy cyclones and wind for rainy cyclones in Figs. 7–9*

We think it is needed to provide the full characterization of the different cyclones. Just because a cyclone has a wind extreme, does not mean that there is no precipitation. We think it is important to highlight these differences to show if different categories are unique or not.

*l. 327 Fig. 8 is already referred to on l. 308*

The sentence with the introduction of Fig. 8 in l 308 should not be there and will be removed.

*l. 329 This is true for panels (a) and (d) but not (b) for instance*

We should highlight that this is only the case for wind speed, and not for precipitation. Also, the other reviewer highlighted that selecting precipitation at t0 is problematic, and this method will change. So this section will likely change quite significantly.

*l330, I do not understand: there are triangles in the plots, indicating significant differences for EXC100 (also, it should be clarified that the symbols indicate statistical significance)*

The markers indicate statistical significance for EXC10 (circle), EXC100 (triangle) and EXC1000 (square) between the two regions for each timestep. The definition of the symbols is in the caption of Fig. 8.

*l. 335 This contradicts l. 320 (see comment above)*

See the comment for l 329.

*l. 337 This questions the relevance of the definition of EXCs (see above comments on methods*

See the comment for l 329.

*l. 349 typo: "cyclones"*

Will be changed.

*l. 352–362 Why look at summer cyclones separately? This is quite a long description for a figure that is not shown in the paper*

As mentioned in the review paper by Flaounas et al. (2022), there are not a lot of papers on extreme cyclones in summer and this is a research gap we deem highly significant to address. However, for the sake of space, we decided to put these figures in the supplement.

*l. 365 See comment on l. 303*

See the comment for l. 303.

*l. 380 why "often"? Why "again"?*

"often" and "again" can be removed.

*l. 422ff references to specific figures are unexpected in the conclusion*

This is fair, the references will be removed.

*l. 439 Please explicit, and clarify whether it is your result or arises from the cited study*

The latter part refers to the cited study. The sentence should be changed to something like this:

" even though the horizontal resolution of the climate model used is too coarse, as discussed in  Flaounas et al. (2013)."

*l. 442 the wind and precipitation are also underestimated compared to ERA5*

That is a very good point to highlight!

*l. 451 an earlier study cannot confirm your current results: the other way round*

Thanks, this, will be changed to "Our results confirm this".

*l. 470 Flaounas et al. (2015b)*

Will be changed.

*l. 471 and a very different time period!*

This will be included.

*l. 472 "region": better "area" to avoid confusion with the east/west Med*

Will be changed.

*l. 476 max???*

This was a typo andwill be removed.

*l. 477 Homar et al. (2007)*

Will be changed.

*l. 484 5% in frequency?*

See comment on l. 8. This should be "in the order of 5% from the multi-millennial mean".

Still "up to" should be replaced by roughly, as the multi-decadal variability is roughly 5% for several aspects relating to Mediterranean cyclones.

**References**

Cavicchia, L., Von Storch, H., & Gualdi, S. (2013). A long-term climatology of medicanes. *Climate Dynamics*, *43*(5–6), 1183–1195. https://doi.org/10.1007/s00382-013-1893-7

---

## Author Response (AR1)

**We thank the reviewers for their constructive and helpful comments, which provided a more robust approach and helped us to increase the presentation of our manuscript.**

**Reviewer 1:**

**Major comments**

1: *Given a baroclinic growing cyclone, I would expect maximum precipitation (rates) on average around the time of maximum intensification (e.g. Papritz et al., 2021). This is also reflected in the precipitation composites in e.g. Figure 8, where the maximum precipitation for the extreme wind composites (Fig. b and c) is before the minimum pressure is reached. That the precipitation maximum is still at the minimum pressure for the extreme precipitation composites, is probably due to the sampling on the maximum precipitation, as the authors also remark themselves. That the cyclones grow on average baroclinically is visible in some of the composites a westward tilt of the minimum pressure with height. Can the authors argue why they did made this choice?*

We agree with this comment that it is well-known that cyclone-related precipitation peaks before the cyclone reaches its mature stage (i.e. when the cyclone reaches its lower core pressure) (Booth et al., 2018, Papritz et al., 2021). We decided to look at precipitation at the time of lowest core pressure to make it consistent with wind speed. However, we do agree that this is not the best way to select extreme precipitation cyclones. Therefore, we repeated the analysis for extreme precipitation cyclones by selecting those with the highest precipitation rate independent of the core pressure. Still, the cyclone state with the highest precipitation rate must be located within one of the corresponding boxes of Fig. 1, defining the central and eastern Mediterranean. The cyclone-related precipitation now clearly peaks before the cyclone reaches its minimum pressure on average, as elaborated in section 3.5 and Fig. 8. The reviewer's suggestion greatly increased the relevance of our analysis and we thank the reviewer for the suggestion.

2: *The minimum pressure is not the best way to assess cyclone strength, since it depends on both the latitude (since there is a equator to pole gradient of pressure) and the size, since a minimum pressure does not necessary indicate a strong gradient of pressure. Can the authors argue why this would be a sensible choice, and have they tested for example using the gradient of gpm per 1000 km, since given the cyclone detection algorithm they have this information available?*

We fully agree with the reviewer and can see that the reviewer might have misinterpreted our results. We use wind speed as an indication of strength. We only use minimum core SLP to illustrate the life cycle and to determine the mature stage of a cyclone. In Fig. 8 d,h, we show the minimum core SLP to allow the community to compare our results with existing literature, as still most of the studies use minimum core SLP as a strength measure.
Note that the strength of the cyclone in Fig. 8 in the unrevised manuscript was based on its related cyclone-related precipitation and wind speed at the time of minimal core pressure. As shown by Pfahl and Sprenger (2016), this works well for wind speed. For precipitation, this method has drawbacks, as explained in the reply above, and has been adjusted.

As a side remark, the issue of minimum core SLP versus gradient might not be so relevant here in this study, as our region does not span over a large range of latitudes. We show the gradient–pressure relationship to illustrate this here in the response (see below Fig. S1).
To adapt this point, we revised the section 3.5 accordingly.

[Figure]

*Fig S1: Core pressure (blue) versus gradient (red) for the 10 (striped lines), 100 (normal lines) and 1000 (dotted lines) most extreme cyclones in the central Mediterranean (left) and the eastern Mediterranean (right).*

In Fig. S1, we show the relation between gradient and core pressure for EXC10, EXC100 and EXC1000, similar to Fig. 8g–i in the manuscript. The relation that the highest gradient is achieved when the core cyclone pressure is the lowest is evident for the central Mediterranean and is also true most of the time in the eastern Mediterranean. We assume this will be the same for other measures like cyclone depth. Hence for individual tracks, using the time of minimum core pressure is a good indication of cyclone maturity.

3: *In several parts of the paper (e.g. line 50) the authors argue that the modes of variability influence the occurrence and strength of Mediterranean cyclones. This would suggest some causal link. However, I would argue that it is merely a correlation, as the authors also write for e.g. the link to the NAO (line 48). I would therefore suggest rewriting the text at these points a bit.*

This is a fair point. We agree that our wording was too strong and that the general atmospheric circulation modes are not a cause of cyclones but merely a correlation. These points have been addressed.

**Minor comments**

*Line 8: Does this variability refer to the variability of cyclone frequency?*

We agree that this was not fully clear. We refer to a set of cyclone characteristics, not just frequency. The variability of cyclone frequency, 90th percentile cyclone-related precipitation and 90th percentile cyclone-related wind speed are in the order of 5%.

To clarify this statement, we have changed the sentence to:

L8-9: "We found that Mediterranean cyclone characteristics exhibit pronounced multi-decadal variability in the order of 5% throughout the entire late Holocene with respect to several cyclone-related properties."

*Lines 47-49: This is an almost exact repetition of the sentence before, so I would suggest rewriting it and remove one the lines. I think this sentence is clearer than the previous, so I would suggest keeping this sentence.*

We agree. The first part of the sentence has been removed and the latter part of the sentence about precipitation in the Mediterranean has been placed after the second sentence.

L58-61: "A positive NAO phase is statistically connected with a decrease in cyclone frequency over the Mediterranean, whereas an increase in cyclone frequency is found during the negative phase of the NAO (Raible et al., 2007). Moreover, the NAO is also inversely correlated to wintertime precipitation over the Mediterranean (Brandimarte et al., 2011; Montaldo and Sarigu, 2017)."

Yes, "they" refers to proxies. This sentence has been rewritten to make it flow better:

L82-84: "Yet, since proxies are usually only sensitive to temperature and precipitation, and not to wind and pressure, it is complicated to reconstruct cyclone activity directly (Raible et al., 2021)."

*Line 92: The focus on extremes could be motivated a bit more since it is a vital part of the manuscript.*

We elaborated on some of the literature provided earlier in the first paragraph to show the relation of cyclone presence to certain extremes in the Mediterranean.

L24-27: "Pfahl and Wernli (2012) showed that precipitation extremes and the presence of a cyclone are related up to 80% of the time in the Mediterranean, whereas Nissen et al. (2010) showed that the majority of wind extremes in the Mediterranean is caused by cyclones present in the same region. Portal et al. (2024) showed that compounding rain-wind and wind-wave extreme events in the Mediterranean are mostly caused by cyclones."

*Line 103: 1.9 x 2.5 degree longitude and latitude respectively?*

Yes, indeed, and this has been clarified in the text.

L117: "1.9°(latitude) × 2.5° (longitude)"

*Line 119: Given that you use the Z1000, you probably detect minima in the geopotential (height) and not pressure directly?*

Indeed, the sentence has been changed a bit to make it more consistent:

L133: "The algorithm identified local minima in the Z1000 field."

*Line 121: Local minimum or averaged over the 1000 km?*

It should be the mean gradient averaged over 1000 km. This has been changed to:

L135: "A minimum mean gradient of at least 20 geopotential meters (gpm) per 1000 km is achieved."

*Line 124-125: I do not understand it completely, since as far as I understand the authors track the cyclones on a 6-hourly resolution, so why is this criterion applied daily?*

It means that the new cyclone centre cannot be more than 1000 km away from the previous centre if the time difference between the two centres is 24hr. However, since we have 6-hourly data, this basically means that the new cyclone centre cannot be further away than 250 km. We agree this point was not very clear.his has been clarified in the text as follows:

L139: "The new minimum of the cyclone must be within 250 km of the previous cyclone minimum."

*Line 143: Are these eruptions used, independently, where they occurred on earth?*

Yes, they can occur anywhere on Earth. Most of the major eruptions have a tropical origin and thus can affect both hemispheres. An extra sentence has been added:

L154-155: "These eruptions occurred over the whole globe, however, most occurred in the tropics (16 out of 20)."

*Line 144: Why is there a time frame of 5 years used before an eruption and only a time frame of 2 years after the eruption?*

The time frame of 5 years before the eruptions will provide enough years to characterize internal climate variability (note that we do this for 20 eruptions, which equals 100 years of "undisturbed climate variability"). We only use the first two years after an eruption as these are the years when the impact of the volcanic forcing is strongest, so we expect to see the strongest signals on climate (Robock, 2000). Our results already indicate that the impact is not strong, so extending it to longer periods after an eruption will decrease the signal-to-noise ratio.

*Line 154: This region could be indicated in one of the Figures.*

The region used for the EOF can be found in Fig. S1 in the supplementary file where all the 4 shapes of the EOFs are provided.

*Line 167: See above major remark, why not selecting on the time step of maximum intensification, since this is one would expect strongest precipitation rates?*

See our response to major comment 1.

*Line 176: How many cyclones are detected in total, or in other words, what is the fraction of selected 'extreme' cyclones?*

That is a fair point, and the numbers are 239476 for the central Mediterranean and 161768 for the eastern Mediterranean. These numbershave been included in the text (L191).

*Lines 200-205: The authors argue that the storm tracks are too zonal compared to ERA5 in the CESM model. However, if I would for example look at the DJF climatology, I would almost argue the opposite: there are relatively more cyclones detected at the northern side of the storm tracks, and less at the southern side. Can the authors explain why they argue that the storm tracks are too zonal in CESM?*

The comment that CESM is too zonal mainly refers to the Atlantic, where, in our opinion, the lack of cyclones penetrating the subtropics is evident, and we allocate this to a zonal bias. However, as the other reviewer pointed out, this is probably also a consequence of a northern shift in the storm tracks. This has been accounted for:

L218-219: "Thus, the biases generally indicate that simulated storm track in the North Atlantic is too zonal and shifted northward compared to the storm track in ERA5."

*Line 210: See previous remark, isn't it more a northward shift of the storm tracks (away from the Mediterranean)?*

See our response to the previous comment for Line 200-205, but we nuanced our statement a little.

L224-225: "This could be related to the zonal and northward bias of the storm tracks in CESM, and therefore, cyclones from the Atlantic penetrate the Mediterranean too little."

*Line 229: I might have missed this, but the 850 hPa temperature related to the cyclones is calculated in a certain area/radius around the cyclones? And the plotted temperature in Figure 4 is then the average over all cyclones occurring in a certain year?*

Fig 4a refers to the 30-year running mean T850 anomalies for the two boxes in Fig 1b combined. Fig 4a is not related to any cyclone-related metrics and just acts as a proxy for the state of the climate of the last 3500 years. We have written this more explicitly now.

L241-243: "To show how internal variability and past climate changes affect Mediterranean cyclones, we show 30-year running means of T850 averaged over all grid points of the two boxes (not cyclone related) and several cyclone-related features over the period 1500 BCE to 1850 CE for the two regions combined (Fig. 4)."

*Line 259: See remark above, I would be careful to describe these modes as 'drivers of the circulation'.*

See the response to major comment 3.

"important driver" has been changed to "another important influence" in L272.

*Line 268: See previous remark*

See the response to major comment 3. Also here we nuanced our statement.

L279: "The NAO shows the highest correlation with cyclone frequency over the North Atlantic and central Europe (Fig. 6a)."

*Line 287-288: I think this sentence could be moved to the methodology section, since it is not related to results shown.*

Given that we find no relevant correlation at all and the fact that this sentence does not fit into the rest of the narrative, we decided to remove this sentence from the manuscript.

*Line 300: To what does the three different EXC types refer too? I assume it is the extreme precipitation, wind and compound composites? This could be further clarified.*

Although the commented points are mentioned in section 2.5, we agree it is useful to include them here again for the sake of clarity. The EXC types are included in the revised version.

L297-299: "Although these composites correspond to three distinct EXC types (precipitation, wind speed and compounding events), it is important to acknowledge that an individual EXC may appear within the top 100 rankings across multiple metrics (Table S2)"

*Lines 320-321: I do not understand what the authors try to argue here, can the authors clarify their argument here?*

When observing the wind speed composites Fig. 7, visually, there are wind speed and compounding EXCs that are much stronger in the central Mediterranean compared to the eastern Mediterranean (Fig. 7b vs 7e and Fig. 7c vs 7f). This is also evident from the dots indicating statistical differences between the two. The argument here is that grid cells with statistical significance have a much more coherent pattern for the wind speed composites (which mostly coincide with the areas with the highest wind speeds) than the precipitation composites. The statistical significance pattern for the precipitation compounds is much less coherent. We understand, however, where the confusion comes from and we tried to clarify this.

L316-317: "Differences between the central and eastern Mediterranean are statistically significant (5% level) for precipitation EXCs, but compared to wind speed EXCs, the grid cells that are significantly different are more randomly dispersed (stippling in Fig. 7)."

*Lines 337-338: Given this possible pre-selection bias, I would strongly suggest the authors to look at the sensitivity of this choice.*

See reaction to major comment 1.

*Line 352: Is Figure 8 then DJF?*

Yes, we added DJF to the first sentence discussing Fig. 8 (L324)

*Line 369: The authors write that the jet stream remains at certain position, which suggests that the jet stream remains at the certain position over a time period, but I don't think that is what the authors mean.*

This is indeed not what we mean, and we appreciate the reviewer's sharp observation. What is meant here is that the jet stream is located southeast of the cyclone during the mature phase. Hence, this has been changed to the following:

L363-364: "Also, the jet stream is located southeast of the EXC100 centre in the eastern Mediterranean, whereas the jet stream in the central Mediterranean is only located south and southwest of the EXC100 centre."

*Line 378: This already suggests that the (detected) cyclones grow baroclinically*

Thank you for the suggestion. This has been included in the sentence.

L371-372: "In Fig. 9g–l, the cyclone centres are often located just east of the lowest Z500 anomalies and the maximum RWP amplitude, highlighting a westward tilting of the cyclone in the upper atmosphere that indicates that these cyclones grow baroclinically."

*Line 385: What is meant with an areawise more negative anomaly?*

This simply means that EXCs in the central Mediterranean are accompanied by a larger negative Z500 anomaly with respect to size (i.e. a larger trough in size). This has been clarified.

L378-379: "EXC100s in the central Mediterranean are accompanied by a spatially larger negative Z500 anomaly than in the eastern Mediterranean."

*Lines 459-465: I would suggest writing the abbreviation EXC in full, since this probably would clarify the text.*

We agree and this has been changed to "ourextreme cyclone (EXC) composite…" (L453).

*Caption Figure 4c: I think it is a precipitation rate (in mm/6h), as also described in the label of the y-axis?*

Yes, this is an error and has been corrected.

*Figure 7 and elsewhere: I would suggest to make the text of the regions Central and Eastern Mediterranean bold, the first time I read the figure labels I was confused because I read them as 'central Mediterranean longitude'*

We thank the reviewer for the suggestion, and we have made the labels more distinct.

**Reviewer 2**

**We thank the reviewer for their constructive and helpful comments, which provided a more robust approach and helped us to increase the presentation of our manuscript.**

*1: The structure of the results section is imbalanced. On the one hand, the first half presents an overview of different aspects, where many results apply to the North Atlantic rather than the Mediterranean (Figs. 2, 5, 6). They may be relevant for the Mediterranean but this is not discussed. For instance, not much impact of volcanic eruptions is found, nor much link with atmospheric modes. On the other hand, the second half describes intense cases only. It is very detailed but contained in a single section. Some reorganization is required here, with important results highlighted and others streamlined.*

We thank the reviewer for their suggestion. However, we do not fully agree with this comment. One of our intentions of including the Atlantic is because the region is upstream to the Mediterranean. In addition, we would like to argue that the Atlantic offers a possibility of a comparison of the role of volcanic eruptions and atmospheric modes of circulation in the Mediterranean.
Responding to the reviewer's comment, we can see that the arguments we attempted to deliver were not clear. So, we have adjusted the text to motivate each section better and explain the relevance of the Mediterranean.
We also agree that some sections need a smoother transition and some figures show too much information. For Figs 2 and 3, we only show the results for DJF and JJA. As MAM and SON are not discussed in other sections of the paper, we removed these results from the manuscript. Furthermore, we also removed the middle and right columns of Fig. 6 of the old manuscript, as the discussed modes of circulation have no significant impact on cyclone-related wind speed and precipitation in the Mediterranean. Lastly, we also divided the Results section into two separate sections. The first section discussing long term variability of Mediterranean cyclones, and the second section discussing extreme Mediterranean cyclones.

*2: Why are results compared between the western/central and eastern Mediterranean? Dynamical differences between these regions are not introduced, despite the large body of literature about Mediterranean cyclones. In contrast, Genoa lows, Vb cyclones, Sharav cyclones and Medicanes are introduced but not further discussed. Thus, the reader does not know what to learn from the comparison.*

We thank the reviewer for raising this point. We do not justify enough why we compare the two regions. Partly, this is due to the differences we found in the results, but we agree that we have to justify this in the introduction. Therefore, we added the papers by Trigo et al. (2002) and Doiteau et al. (2024) to highlight the difference in the mean climate and circulation characteristics between the subregions. We especially thank the reviewer for providing the latter reference.

*3: Why is CESM compared with ERA5 for the period 1980–2010 only and not for the full period 1940–present?*

The 30-year reference periods are used to define climate averages as defined by the WMO. Also, the time period suggested by the reviewer overlaps with a strong increase in global temperature and its potential effects on global circulation. Additionally, the pre-1979 ERA5 satellite observations were sparse, which can cause the risk of including biased data with respect to the storm tracks. Note also that extending the period to the present would mean that we compare observations with emission scenario-driven climate (CESM). Therefore, we see no reason to change this time period.

4. Methods are not always clear and definitions are sometimes repeated, which blurs the interpretation of results.

We agree that some definitions in the methods section are defined vaguely or repeated unnecessarily. With the reviewer's suggestions in the minor comments below, we have addressed them and tried to make sure the methods section overall becomes clearer and easier to understand.

5. In the results section, please systematically refer to the figure and panel that is discussed. Currently, one has to guess where to look at. Also, figures tend to contain many panels, which are not all discussed and could likely be removed to focus on the main results.

We agree that a lot of panels are not properly referred to, especially in the last few figures - we thank the reviewer for pointing them out. Additionally, in the reply to major comment 1, we implemented changes to Figs. 2,3 and 6 to make the information conveyed in the paper as a whole more concise.

**Minor comments**

*l. 3 "their variability in the late Holocene is poorly understood": more precisely?*

More precisely, their spatial and temporal variability in the late Holocene. This has been changed to:

L3-4: "their spatial and temporal variability is poorly understood"

*l. 8 5% in what?*

The variability of cyclone frequency, 90th percentile cyclone-related precipitation and 90th percentile cyclone-related wind speed are in the order of 5%.

This has been changed to:

L8-9: "We found that Mediterranean cyclones exhibit pronounced multi-decadal variability in the order of 5% throughout the entire late Holocene with respect to several cyclone-related properties."

*l. 9 the relation is described as "weak" in the conclusions*

L9-10: We added "weak statistical relation" in the abstract to clarify.

*l. 24–25 What kind of variability and connection? This is the main motivation for the paper, thus requires (way) more details. Perhaps it is discussed below but it is unclear at that point.*

We agree this sentence is unclear. We have rewritten it in the following way:

L28-29: "However, the factors driving the variability of Mediterranean cyclone characteristics, especially extreme cyclones, are not fully understood."

*l. 40–41 References are expected here*

The paper by Cavicchia et al. (2013) has been included to refer to the Medicane climatology, and the sentence has been changed to the following (including that the peak also takes place in winter:

L52-54: "A special type of Mediterranean cyclone are so-called Medicanes, which are hybrid systems between tropical and extratropical storms and often occur in autumn and winter (Cavicchia et al., 2013)"

*l. 42 I don't fully agree: see, e.g., the devastating cyclone Daniel of September 2023*

This is a fair point, but we argue that extratropical cyclones are much more likely to occur in the Mediterranean, and their cumulative impact is much larger. We weakened the statement to:

L54-55: "However, due to their rarity, their overall socio-economic impact is not as large as that of extratropical Mediterranean cyclones (Flaounas et al., 2022)."

*l. 43–44 Feser et al discuss North Atlantic cyclones whereas Flaounas et al discuss Mediterranean cyclones*

The reference of Feser et al. (2015) is about decadal variability observed in extratropical cyclones in NW Europe and the Atlantic. The reference of Flaounas et al. (2022) is with respect to section 2.2, where the relation between Mediterranean cyclones and teleconnection patterns is discussed. However, discussing the relation for extratropical cyclones in a more general sense makes more sense. Therefore, instead of Flaounas et al. (2022), we have included the references of Seierstad et al. (2007) and Walz et al. (2018) in the text.

L62-66: "Other modes, like the East Atlantic (EA), Scandinavian (SCAN) and East Atlantic Western Russian (EAWR) and the Polar-Eurasian (POL) modes of variability, also have a significant influence on cyclones in the Northern Hemisphere (Seierstad et al., 2007). In addition to the modes of atmospheric circulation, Walz et al. (2018) hypothesized that sea ice anomalies are a dominant factor in the inter-annual variability of Mediterranean cyclones."

*l. 46 Wintertime precipitation correlates negatively with NAO but positively with cyclone frequency*

This is an error in the structure of the sentence. The sentence has been changed to:

L58-61: "A positive NAO phase is statistically connected with a decrease in cyclone frequency over the Mediterranean, whereas an increase in cyclone frequency is found during the negative phase of the NAO (Raible et al., 2007). Moreover, the NAO is also inversely correlated to wintertime precipitation over the Mediterranean (Brandimarte et al., 2011; Montaldo and Sarigu, 2017)."

*l. 65 What scales are referred to by short/long periods?*

This should be of a smaller spatial scale and has been changed to:

L77-79: "Mediterranean cyclones, which are usually of a smaller spatial scale than other extratropical cyclones, suffer even more from the low resolution in GCMs (Flaounas et al., 2013)"

*l. 84 What scale is referred to by low frequency?*

With low-frequency, we referred to multi-decadal time scales. Low frequency has been replaced in the text with the latte (L97).

*l. 88 It would be worth citing and discussing the few studies, as the impact of volcanic eruptions on cyclones is investigated in the paper*

There are very few studies directly investigating the link between volcanic eruptions, and the ones we found are mentioned in the two sentences before. Though with some extra research, we did find an earlier study by Fischer-Bruns et al. (2005), who found no link between volcanic eruptions and storm activity. This study has been included in the introduction as well.

L101-102: "However, earlier work by Fischer-Bruns et al. (2005) found no relation between differences in volcanic forcing and storm activity, and thus the effect of volcanic eruptions on cyclones remains uncertain.

*l. 100 missing "in" Kim et al*

This sentence has been changed to:

L124: "How these forcings were obtained and generated are explained in detail in Kim et al. (2021)."

*l. 115 why not 1940–2024?*

See response to major comment 3.

*l. 125 I don't fully understand the condition "across" 24 hours*

It means the new cyclone centre cannot be more than 1000 km away from the previous centre if the time difference were to be 24h. However, since we have 6-hourly data, this basically means that the new cyclone centre cannot be further away than 250 km. This has been clarified in the text as follows:

L139: "The new minimum of the cyclone must be within 250 km of the previous cyclone minimum."

*l. 134 missing "in" Raible et al*

L149: Brackets have been included "(Raible et al., 2018)."

*l. 134–138 What is the meaning of fitting a Gaussian function to the geopotential field on a single grid point? Also, rather than describing first the "traditional" selection and then the adaptation, I recommend describing straight away what is actually used here.*

We agree that this part is a bit confusing, and we want to highlight that we do not fit a Gaussian function to one grid cell, as that would be impossible. What we meant is that sometimes the radii are so small that they do not extend over more than one grid cell. Still, to clarify, we have rewritten the last paragraphs of section 2.2 as follows (shortening it significantly):

L141-150: "The cyclone identification and tracking method provided a variety of cyclone characteristics, such as the cyclone position, the radius of a cyclone, the cyclone depth, the core (central) pressure, and cyclone-related mean and extreme precipitation and wind speed. To compute the cyclone radius, first, a Gaussian function was fitted to the Z1000 field, assuming that the cyclone was azimuthally symmetric (Schneidereit et al., 2010). The cyclone radius is then defined as the distance between the cyclone centre and the point of 1.5 standard deviation (which represents the middle between the first and second inflection points), as done by Messmer and Simmonds (2021). The depth of the cyclone is defined as the difference between Z1000 in the centre of the cyclone and the Z1000 mean over the area of 1000 × 1000 km2. To calculate the cyclone-related wind speed and precipitation, the maximum value of wind speed and precipitation of all grid cells within a cyclone's radius

were considered (Raible et al., 2018). We also applied the tracking algorithm to the ERA5 Z1000 field to evaluate the Mediterranean cyclone characteristics in CESM, mostly focusing on the cyclone tracks."

L159: It has been corrected.

L170: Has been corrected.

*l. 161–165 not sure what "western Mediterranean" means here; actually the CMED region in Fig. 1 largely contains the region usually referred to as the West Med. Thus, for consistency with previous studies, and considering that only two regions are compared here, I recommend naming them West/East Med.*

As our box for the central Mediterranean is not far enough west to cover the full western Mediterranean, we decided to keep our definitions of the central and eastern Mediterranean as they were.

*l. 166–177 The precipitation and wind speed metrics are repeated several times in the paragraph but in an inconsistent fashion (with/out max, with/out t0); I recommend defining them once for good and then simply using precipitation and wind speed (as in l. 177). Also, which radius is used along the track for precipitation and wind speed?*

According to the reviewer's comment, we updated the paragraph as follows:

L178-181: "To do this, we ranked them based on WS850 (hereafter just wind speed) at the time when the cyclone reaches the lowest core sea level pressure within its track (hereafter t0) (similar to Pfahl and Sprenger (2016)), and based on the maximum 6-hourly cyclone-related precipitation (hereafter just precipitation) within the entire track that affects the region."

The rest of the manuscript has been corrected accordingly.

For the selection of extreme precipitation cyclones, the assumption is that one picks the cyclones from the tail of a Gaussian distribution. This clearly is not the case for precipitation, and therefore, we compute the square root of the cyclone-related precipitation so the distribution looks more Gaussian.

To clarify this, we explained this in more detail in the manuscript:

L185-187: "The distribution for precipitation is skewed towards extreme values and values close to zero. Therefore, all precipitation rates below 1 mm 6h-1 were excluded, and we computed the square root of the remaining precipitation to approximate a Gaussian distribution for precipitation."

*l. 177 here and elsewhere: CMED and EMED to keep the same terminology*

See comment for l. 161-165.

*l. 180 t0 already defined*

We agree, and the sentence has been changed to the following:

L195-196: "For each cyclone track, we set the reference time t0. Every time step of the track that occurred after t0 obtains a positive index, and the time steps of the track that occurred before t0 receive a negative index."

*l. 185 missing "at" t0?*

Indeed missing "at", has been included (L199).

*l. 186 variations in what? I don't get the point here*

This refers to the "variations of the weights of the grid cell". Due to the small range of latitudes and the southerly location of the Mediterranean, we deem the differences in weights of the grid cells insignificant for the EXC composites. However, the sentence should be rephrased to make it clearer:

L199-201: "Since all the cyclones were located in the Mediterranean basin at t0 and, therefore, within a relatively small range of latitudes, variations due to differences in latitude weights of grid cells were considered insignificant and, therefore, negligible."

*l. 188 Why 30h? And missing "for" t0?*

30h is an arbitrary value. We deem it enough to capture the intensification and decaying phase of the cyclone. We also tried plotting 48h before and after the cyclone, but it did not provide any extra insights. And indeed, "for" is missing and has been added.

*l. 191 are all fields averaged over the area of both regions for all cyclones?*

Bullet point number 4 should be clearer. We do not mean we compute averages of all fields that we have. We computed spatial averages of the fields noted in the paragraph below. To clarify this, we changed bullet point 4 to the following:

"For every time step within these 30 hours before and after t0, we compute spatial averages for EXC10, EXC100, and EXC1000. This was done for both regions for cyclones associated with cyclone-related precipitation, wind, and compound extremes. Cyclone tracks, which did not appear at any of these time steps, were ignored for the temporal means."

*l. 192 Cyclone tracks "that" did not appear (and no comma)*

This has been changed (L206).

*l. 195 Some details are expected about the RWP amplitude. And typo: Rossby wave "packet"*

More information on the method has been included, and the typo has been corrected.

L207-209: "The composite analysis was applied to the precipitation, wind speed, WS300, SLP, T850, Z500 and the Rossby wave packet (RWP) amplitude. The computation of the RWP amplitude is based on the method provided by Fragkoulidis et al. (2018) and is produced by calculating the amplitude of the envelope of the 300 hPa meridional wind"

*l. 200ff In the discussion on Fig. 2 please indicate which panel is referred to*

As said in major comment 5, we have tried to refer to the figure panels more consistently throughout the text.

*l. 204 Fig. 2 suggests a northward shift rather than a zonal vs wavy storm track*

We agree that Fig. 2 also shows a northward shift of the jet. However, it is not just a northward shift of the jet, but also a decrease in waviness (e.g. CESM not capturing cyclone activity around the Azores in DJF). The sentence has been changed to:

L218-219: "Thus, the biases generally indicate that the simulated storm track in the North Atlantic is too zonal and shifted northward compared to the storm track in ERA5"

*l. 209–214 Any evidence for these assumptions? Otherwise they sound speculative*

In the introduction (line 31), we mention that about 20% of Mediterranean cyclones originate in the Atlantic. Now, although this does not account for the 50% underestimation we see in Fig. 2, we hypothesize that a less wavy and more northerly jet stream would contribute to fewer cyclones penetrating the Mediterranean and would also hamper conditions for lee cyclogenesis. We argue this is not merely speculation.

*l. 216 "while" rather than "although"*

This has been changed (L229)

*l. 221 this is speculative but should be easy to verify*

To avoid speculation, we removed the sentence hypothesizing about the heat lows.

*l. 224 where is it most often underestimated by 50%? it is unclear where this number comes from*

This refers to the tails of the distribution of Fig. 6m–p, and the difference is about 50%. To clarify, we have rewritten it to the following:

"CESM struggles to reproduce high cyclone-related precipitation events in all four seasons, where the high-end tails of the distributions are most often underestimated by 50%."

*Fig. 3 wind speed is the max value here, while it is defined as the value at t0 in Section 2.5*

We appreciate the reviewer's sharpness here. We agree that it is not very consistent to use max wind speed in a cyclone track instead of the wind speed at t0, so this has been adjusted in Fig. 3i–j. However, we do emphasize that we do not see big differences here.

*l. 234 the 30y running mean does not show similar patterns and is not appropriate for tendencies*

Our statement in line 234 is not worded properly and, therefore, causes confusion. What we mean is that in both Fig. 4a and Fig. 4b multidecadal variability is clearly visible, not that the variability in Fig. 4a and 4b are correlated. To clarify this, we changed line 234 to the following:

L249: "The cyclone frequency in the Mediterranean (Fig. 4b) also exhibits a clear multidecadal variability."

*l. 235 is this definition of cyclone frequency different from that used elsewhere in the paper?*

Slightly, as we sum the individual cyclone centres per time step in both regions for every month. However, Fig 4b does not take into account the cyclone radius as in Fig 1. To clarify the differences, we changed the label of Fig 4b to "total cyclone time steps".

*l. 255–256 Repeats previous sentences*

This sentence has been shortened to:

L269-270: "Only a small region with a significant increase in cyclone frequency is found in the eastern Mediterranean, but it still accounts for an increase of 30% in this region."

*l. 260 "important" driver?*

This sentence changed significantly, and thus important is not present in this context anymore.

*l. 263–265 Please refer to the corresponding panels*

Correct references to panels have been included (L276-278).

*l. 266 Fig. 6 is already presented before*

Good point. The first sentence has been removed to make the paragraph flow better.

*l. 287 this sentence is surprising, as sea ice anomalies are not discussed in the methods*

After reviewer #1 also highlighted this, and since this sentence does not really fit in the rest of the manuscript, we have decided to remove it.

*l. 289 this should be mentioned earlier, and clarified that Fig. 6 shows DJF only*

That is a fair point, of course, DJF has been included in the first paragraph of section 3.4

"Since the large-scale variability can be an import driver for Mediterranean cyclones, we show the impact of the four most dominant modes of circulation in the North Atlantic European region, resulting from the EOF-analysis applied to the CESM Z500 fields, on extratropical cyclones in Fig. 6 for DJF."

*l. 303 the presence of the warm sector of the cyclone (I) could be verified and (II) does not dynamically explain the highest wind speeds (see, e.g., Raveh-Rubin and Wernli 2015, or papers for the North Atlantic)*

It is true that the presence of the warm sector is not the cause of the highest wind speeds. However, it is also obvious from Fig. 9 that at t0, the location of the warm sector and the highest wind speeds are correlated. To avoid any confusion or speculation, we removed this statement from the manuscript.

*l. 309–311, 320--321 Please refer to the corresponding panels in Fig. 7*

They have been included

*l. 312–313 not only northward but also (obviously) westward! Western/central and eastern Mediterranean cyclones have different dynamics, which should be discussed in the introduction (see, e.g., Doiteau et al. 2024, or older papers)*

That is a very fair point, and as discussed in major comment 2, we agree there should be more emphasis on the differences between the two regions. Hence, we have expanded the 2nd paragraph of the introduction to accommodate the difference in cyclone dynamics for the different Mediterranean regions. We included the work of Trigo et al. (2002) and Doiteau et al. (2024) in this paragraph. We thank the reviewer for suggesting the latter paper.

*l. 316 it is expected indeed; I don't quite get the point at showing precipitation for windy cyclones and wind for rainy cyclones in Figs. 7–9*

We think it is needed to provide the full characterization of the different cyclones. Just because a cyclone has a wind extreme, does not mean that there is no precipitation. We think it is important to highlight these differences to show if different categories are unique or not.

*l. 327 Fig. 8 is already referred to on l. 308*

In line 308, it should be Fig 7b and has been corrected (L305).

*l. 329 This is true for panels (a) and (d) but not (b) for instance*

We highlight that this is only the case for wind speed and not for precipitation. Also, the other reviewer highlighted that selecting precipitation at t0 is problematic, and this method has changed. So, this section has changed quite significantly.

*l330, I do not understand: there are triangles in the plots, indicating significant differences for EXC100 (also, it should be clarified that the symbols indicate statistical significance)*

The markers indicate statistical significance for EXC10 (circle), EXC100 (triangle) and EXC1000 (square) between the two regions for each time step. The definition of the symbols is in the caption of Fig. 8.

*l. 335 This contradicts l. 320 (see comment above)*

See the comment for l 329.

*l. 337 This questions the relevance of the definition of EXCs (see above comments on methods*

See the comment for l 329.

*l. 349 typo: "cyclones"*

Has been corrected.

*l. 352–362 Why look at summer cyclones separately? This is quite a long description for a figure that is not shown in the paper*

As mentioned in the review paper by Flaounas et al. (2022), there are not a lot of papers on extreme cyclones in summer, and this is a research gap we deem highly significant to address. However, for the sake of space, we decided to move these figures in the supplement.

*l. 365 See comment on l. 303*

See the comment for l. 303.

*l. 380 why "often"? Why "again"?*

"often" and "again" are removed.

*l. 422ff references to specific figures are unexpected in the conclusion*

This is fair, the references have been removed.

*l. 439 Please explicit, and clarify whether it is your result or arises from the cited study*

The latter part refers to the cited study. The latter of the sentence has been changed to:

L433-434: "...even though the horizontal resolution of the climate model used is too coarse, as discussed in Flaounas et al. (2013)."

*l. 442 the wind and precipitation are also underestimated compared to ERA5*

That is a very good point and has been highlighted with an extra sentence:

L435-438: "However, the low resolution of our model may hamper the ability to find relations between cyclone frequency and modes of circulation. Additionally, it should be noted that the cyclone-related precipitation and wind speed in CESM are underestimated compared to ERA5."

*l. 451 an earlier study cannot confirm your current results: the other way round*

This has been changed to:

L445-446: "Our results show no significant relationship between the Mediterranean cyclone variability and solar irradiance, which confirms the aforementioned study."

*l. 470 Flaounas et al. (2015b)*

Has been changed.

*l. 471 and a very different time period!*

The time period that Flaounas et al., 2015b cover has been included

L462-464: "Flaounas et al. (2015b) performed a similar composite analysis of intense cyclones using a regional model with a higher horizontal resolution (20 km horizontal resolution) but considering the Mediterranean basin as a whole for the period 1898–2008."

*l. 472 "region": better "area" to avoid confusion with the east/west Med*

Has been changed.

*l. 476 max???*

This was a typo and has been removed.

*l. 477 Homar et al. (2007)*

Has been changed.

*l. 484 5% in frequency?*

See comment on l. 8. This should be "in the order of 5% from the multi-millennial mean".

Still "up to" should be replaced by roughly, as the multi-decadal variability is roughly 5% for several aspects relating to Mediterranean cyclones.

L475-476: "To conclude, there is no obvious single driver of Mediterranean cyclone variability, Mediterranean cyclones vary on multi-decadal scales with an amplitude around roughly 5% from the multi-millennial mean."

**References**

Cavicchia, L., Von Storch, H., & Gualdi, S. (2013). A long-term climatology of medicanes. *Climate Dynamics*, *43*(5–6), 1183–1195. https://doi.org/10.1007/s00382-013-1893-7

Trigo, I. F., Bigg, G. R., & Davies, T. D. (2002). Climatology of cyclogenesis mechanisms in the Mediterranean. *AMETSOC*. https://doi.org/10.1175/1520-0493(2002)130

Doiteau, B., Pantillon, F., Plu, M., Descamps, L., & Rieutord, T. (2024). Systematic evaluation of the predictability of different Mediterranean cyclone categories. *Weather and Climate Dynamics*, *5*(4), 1409–1427. https://doi.org/10.5194/wcd-5-1409-2024

Fischer-Bruns, I., Von Storch, H., González-Rouco, J. F., & Zorita, E. (2005). Modelling the variability of midlatitude storm activity on decadal to century time scales. *Climate Dynamics*, *25*(5), 461–476. https://doi.org/10.1007/s00382-005-0036-1

Booth, J. F., Naud, C. M., & Jeyaratnam, J. (2018). Extratropical Cyclone Precipitation Life Cycles: A Satellite‐Based Analysis. In Geophysical Research Letters (Vol. 45, Issue 16, pp. 8647–8654). American Geophysical Union (AGU). https://doi.org/10.1029/2018gl078977

Hurrell, J. W. (1995). Decadal trends in the North Atlantic oscillation: regional temperatures and precipitation. *Science*, *269*(5224), 676–679. https://doi.org/10.1126/science.269.5224.676

Papritz, L., F. Aemisegger, and H. Wernli, 2021: Sources and Transport Pathways of Precipitating Waters in Cold-Season Deep North Atlantic Cyclones. *J. Atmos. Sci.*, **78**, 3349–3368, https://doi.org/10.1175/JAS-D-21-0105.1.

Pfahl, S., & Sprenger, M. (2016). On the relationship between extratropical cyclone precipitation and intensity. In Geophysical Research Letters (Vol. 43, Issue 4, pp. 1752–1758). American Geophysical Union (AGU). https://doi.org/10.1002/2016gl068018

Robock, A. (2000). Volcanic eruptions and climate. *Reviews of Geophysics*, *38*(2), 191–219. https://doi.org/10.1029/1998rg000054

---

## Author Response (AR2)

**We thank the reviewer once again for their constructive and helpful comments, which provided the last few corrections for our manuscript.**

*The authors have addressed my main concerns. The paper is now better structured with Section 3 presenting general results and Section 4 focusing on extremes. General knowledge about Mediterranean cyclones is now described in the introduction and definitions are clear and consistent. Also, Section 3.4 has been streamlined to better focus on the Mediterranean. Only I am still not convinced by the interest of discussing volcanic eruptions, at least not in the main paper. They are certainly important but Figure 5 shows their impact is almost non-existent in the Mediterranean.*

As also suggested by the editor, we agree that the effect of volcanic eruptions on Mediterranean cyclonesis not significant enough to be represented in the paper and we have removed all references to volcanic eruptions in the methods, results and discussion.

*l. 186 and elsewhere  should be "10 mm (6 h)$^{-1}$"*

We have included brackets in all units containing mm (6h)$^{-1}$ throughout the entire manuscript.

*l. 191   very exact numbers are not needed*

As these are just the total number of cyclones, and not a fractional number we donot really see the point of rounding them. Therefore, we have decided to leave the exact numbers in the manuscript.

*l. 218   I still do not agree "the simulated storm track in the North Atlantic is too zonal". The above text says the opposite: "CESM overestimates cyclone frequency over the polar North Atlantic and underestimates cyclone frequency in the subtropical North Atlantic"*

We eventually agree with the reviewer that the main pattern we observe related to the storm tracks is mainly driven by a northward shift of the storm tracks and to a lesser extent by a zonal bias in the model. We have rewritten the sentence in the following way:

*"Thus, the biases generally indicate that the simulated storm tracks in the North Atlantic have a northward bias, and are to a lesser extent too zonal, especially in DJF."* (l. 205-206)

*l. 224   see previous comment*

We have changed the sentence as follows:

*"This could be related to the storm track bias in CESM, and therefore, cyclones from the Atlantic penetrate the Mediterranean too little."* (l. 210-211)

Also in l. 213-214 we changed the following sentence:

*"However, the slight overestimation in the eastern Mediterranean suggests that the zonal bias in CESM only plays a role in the central Mediterranean."*

to

*"However, the slight overestimation in the eastern Mediterranean suggests that the storm track bias in CESM only plays a role in the central Mediterranean."*

*l. 426   very locally*

This part will be removed as it relates to the removed analysis of volcanic eruptions.

*l. 431   see above*

See comment to l. 426.

*l. 447   I do not agree: there is almost no impact here either (for the Mediterranean)*

See comment to l. 426.